# Multiparameter prediction of myeloid neoplasia risk

**Muxin Gu**[1,2], **Sruthi Cheloor Kovilakam**[1,2], **William G. Dunn**[1,2],
**Ludovica Marando**[1,2], **Clea Barcena** [1,2,3], **Irina Mohorianu**[1], **Alexandra Smith**[4],
**Siddhartha P. Kar** [5,6,7], **Margarete A. Fabre**[1,2,8], **Moritz Gerstung**[9],
**Catherine A. Cargo**[10,11], **Luca Malcovati** [12,13], **Pedro M. Quiros** [1,2,14] ✉
**& George S. Vassiliou** [1,2,15] ✉

The myeloid neoplasms encompass acute myeloid leukemia, myelodysplastic syndromes and myeloproliferative neoplasms. Most cases arise from the shared ancestor of clonal hematopoiesis (CH). Here we analyze data from 454,340 UK Biobank participants, of whom 1,808 developed a myeloid neoplasm 0–15 years after recruitment. We describe the differences in CH mutational landscapes and hematology/ biochemistry test parameters among individuals that later develop myeloid neoplasms (pre-MN) versus controls, finding that disease-specific changes are detectable years before diagnosis. By analyzing differences between 'pre-MN' and controls, we develop and validate Cox regression models quantifying the risk of progression to each myeloid neoplasm subtype. We construct 'MN-predict', a web application that generates time-dependent predictions with the input of basic blood tests and genetic data. Our study demonstrates that many individuals that develop myeloid neoplasms can be identified years in advance and provides a framework for disease-specific prognostication that will be of substantial use to researchers and physicians.

The myeloid neoplasms encompass the myeloproliferative neoplasms (MPN), myelodysplastic syndromes (MDS), chronic myelomonocytic leukemia (CMML) and acute myeloid leukemia (AML), and collectively affect approximately 10 per 100,000 individuals per year. Advances in understanding their molecular pathogeneses have led to the development of some new therapies; however, the majority of patients with myeloid neoplasms still succumb to their disease[1,2]. Recently, it became clear that in the majority of cases, myeloid neoplasms develop from clonal hematopoiesis (CH), their shared preclinical ancestor[3–6]. We and others have shown that individuals en route to developing AML can be identified years in advance by the genetic characteristics of their CH[7,8], proposing that AML prevention may be a viable alternative to the treatment of established disease[9]. However, our ability to identify those at risk remains limited and is largely derived from targeted case-control studies[7,8].

The study of large cohorts of volunteers has been instrumental in understanding genetic determinants of common and rare diseases[10]

[1]Wellcome-MRC Cambridge Stem Cell Institute, University of Cambridge, Cambridge, UK. [2]Department of Haematology, University of Cambridge, Cambridge, UK. [3]Department of Biochemistry and Molecular Biology, Universidad de Oviedo, Oviedo, Spain. [4]Epidemiology and Cancer Statistics Group, University of York, York, UK. [5]MRC Integrative Epidemiology Unit, University of Bristol, Bristol, UK. [6]Section of Translational Epidemiology, Division of Population Health Sciences, Bristol, Medical School, University of Bristol, Bristol, UK. [7]Early Cancer Institute, Department of Oncology, University of Cambridge, Cambridge, UK. [8]Centre for Genomics Research, Discovery Sciences, BioPharmaceuticals R&D, AstraZeneca, Cambridge, UK. [9]Division of Artificial Intelligence in Oncology, DKFZ, Heidelberg, Germany. [10]Haematological Malignancy Diagnostic Service, St James's Hospital, Leeds, UK. [11]Department of Haematology, Leeds Teaching Hospitals, Leeds, UK. [12]Department of Molecular Medicine, University of Pavia, Pavia, Italy. [13]Department of Hematology, Fondazione IRCCS Policlinico San Matteo, Pavia, Italy. [14]Instituto de Investigación Sanitaria del Principado de Asturias, ISPA, Oviedo, Spain. [15]Department of Haematology, Cambridge University Hospitals NHS Trust, Cambridge, UK. ✉e-mail: pmquiros@ispasturias.es; gsv20@cam.ac.uk

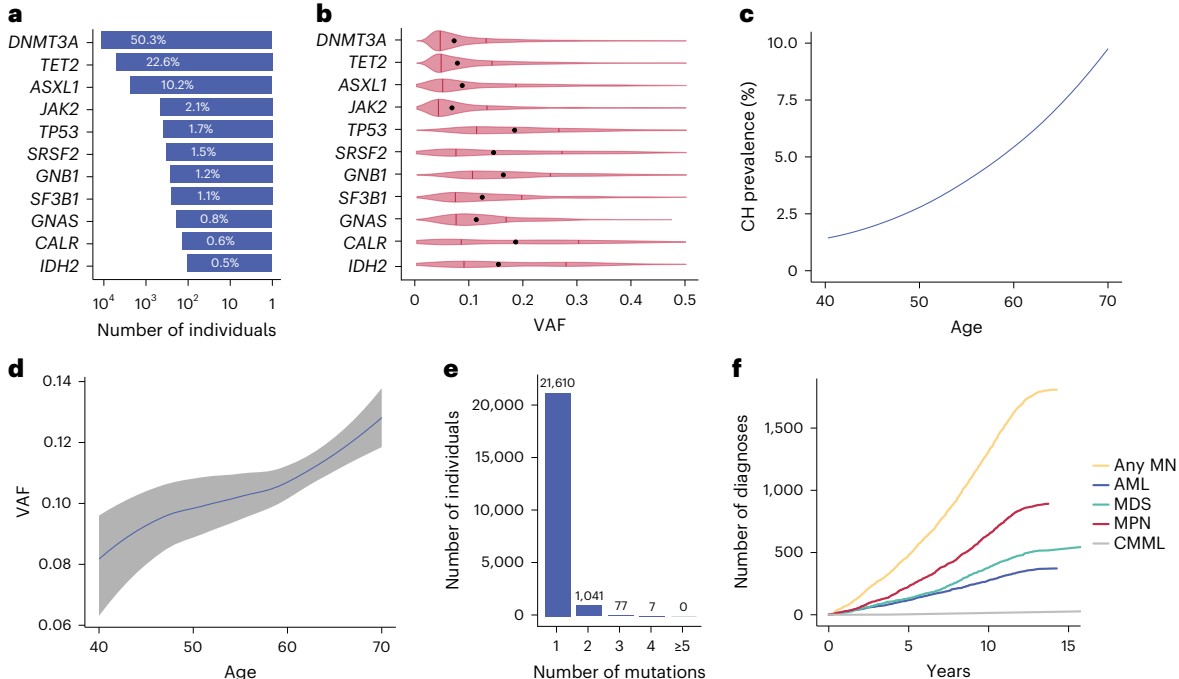

**Fig. 1 | Summary of driver mutations in the 11 most commonly mutated genes in CH. a**, Percentages of cases per driver gene among the 22,735 UKB participants with CH. **b**, Distribution of clone sizes (VAF) by driver mutation. Medians are depicted by black dots and upper/lower quartiles by vertical lines. **c**, Rising prevalence of CH mutations with advancing age. **d**, Increase in size (VAF) of CH clones with advancing age. The line follows the mean of VAFs in each integral age group and the gray area indicates the 5–95% confidence interval estimated by Student's *t*-distribution. LASSO regression was used to smoothen the curves in **c** and **d**. **e**, Number of individuals with 1, 2, 3, 4 and ≥5 driver mutations. **f**, Cumulative incidence of different types of myeloid neoplasms in the UKB.

and many investigators have pursued this approach to study the causes and consequences of CH[11,12]. We recently analyzed data from 200,453 UK Biobank (UKB) participants and found that certain drivers of CH are associated with a greater risk of progression to myeloid neoplasms than others and that some of these higher-risk mutations were associated with more significant changes in blood cell parameters[13]. In light of these findings, the recent release of data from almost their entire cohort offers an opportunity to use the linked genetic and phenotypic data in the UKB to develop an improved approach for predicting the risk of development of myeloid neoplasms in the general population. To this end, here we study data from 454,340 UKB participants and reveal the genomic landscape of individuals that went on to develop myeloid neoplasms, capture the significance of blood cell and biochemical parameters for myeloid neoplasm risk and construct a new regression model that enables prognostication of the risk of progression to different types of myeloid neoplasms. We go on to validate our model in two independent cohorts of patients with clonal cytopenia of undetermined significance (CCUS), the evolutionary stage between CH and myeloid neoplasm, thus confirming the robustness and clinical utility of our approach. Finally, to help clinicians and researchers dealing with patients with CH or clonal cytopenias, we developed 'MN-Predict' a user-friendly web application to generate individualized risk predictions.

## Results

To identify carriers of CH in the UKB, we analyzed whole-exome sequencing (WES) data from all 454,340 participants using Mutect2 (ref. 14) focusing on 38 genes known to be recurrently mutated in CH and myeloid neoplasm and applied filters adapted from a recent study aimed at harmonizing the identification of CH mutations by removing sequencing artifacts and germline variants[15] (Methods and Supplementary Table 1). To overcome low coverage or mapping problems (*U2AF1*)[16], we carried out a targeted analysis of 22 recurrent mutation hotspots

to complement the mutation calls (Methods and Supplementary Tables 2 and 3). Using these criteria, we identified 23,951 CH driver mutations among 22,735 individuals with driver gene prevalence, clonal sizes, number of variants per sample and age distribution in line with previous reports (Fig. 1a–e)[13].

To investigate the relationship between myeloid neoplasm risk and genetic or nongenetic variables, we analyzed data from all 454,340 UKB participants, including age (56.5 ± 8.1 years, mean ± s.d.), sex (female:male (F:M) = 1.18), CH driver mutations, blood test results at recruitment and electronic health records obtained throughout the study (follow-up: 7.4–15.5 years, median 12.6 years). At the time of recruitment, 648 individuals (of whom 233 had CH driver mutations) had been previously diagnosed with a myeloid neoplasm and an additional 108 had, according to the latest diagnostic criteria[17], blood count results that were consistent with a probable diagnosis of polycythemia vera (*n* = 26; hemoglobin concentration (HGB) = 17.9 ± 1.43 g dl⁻¹ and *JAK2*-V617F variant allele fraction (VAF) = 0.38 ± 0.2, mean ± s.d.) or essential thrombocythemia (*n* = 82; platelet count (PLT) = 675 ± 225 × 10⁹ l⁻¹ mean ± s.d., 51 with *JAK2*-V617F, 25 with *CALR* and 6 with *MPL* mutations). These individuals were excluded from subsequent analyses. During follow-up, 1,937 of the remaining 453,584 individuals developed a myeloid neoplasm at a median of 7.9 years from recruitment, including 372 diagnosed with de novo AML, 517 with MDS, 892 with MPN and 27 with CMML (Fig. 1f). CMML cases shared similar mutation patterns to MDS (Supplementary Fig. 1) and were incorporated into the MDS category for subsequent analyses. Those who developed a chronic myeloid neoplasm (that is, MDS, MPN or CMML) and then progressed to AML were considered under their first myeloid neoplasm diagnosis. The remaining 129 individuals were diagnosed with multiple myeloid neoplasms contemporaneously or with AML followed by another myeloid neoplasm. To avoid misclassification, these were classed as 'MN-indeterminate' and excluded from analyses (Methods).

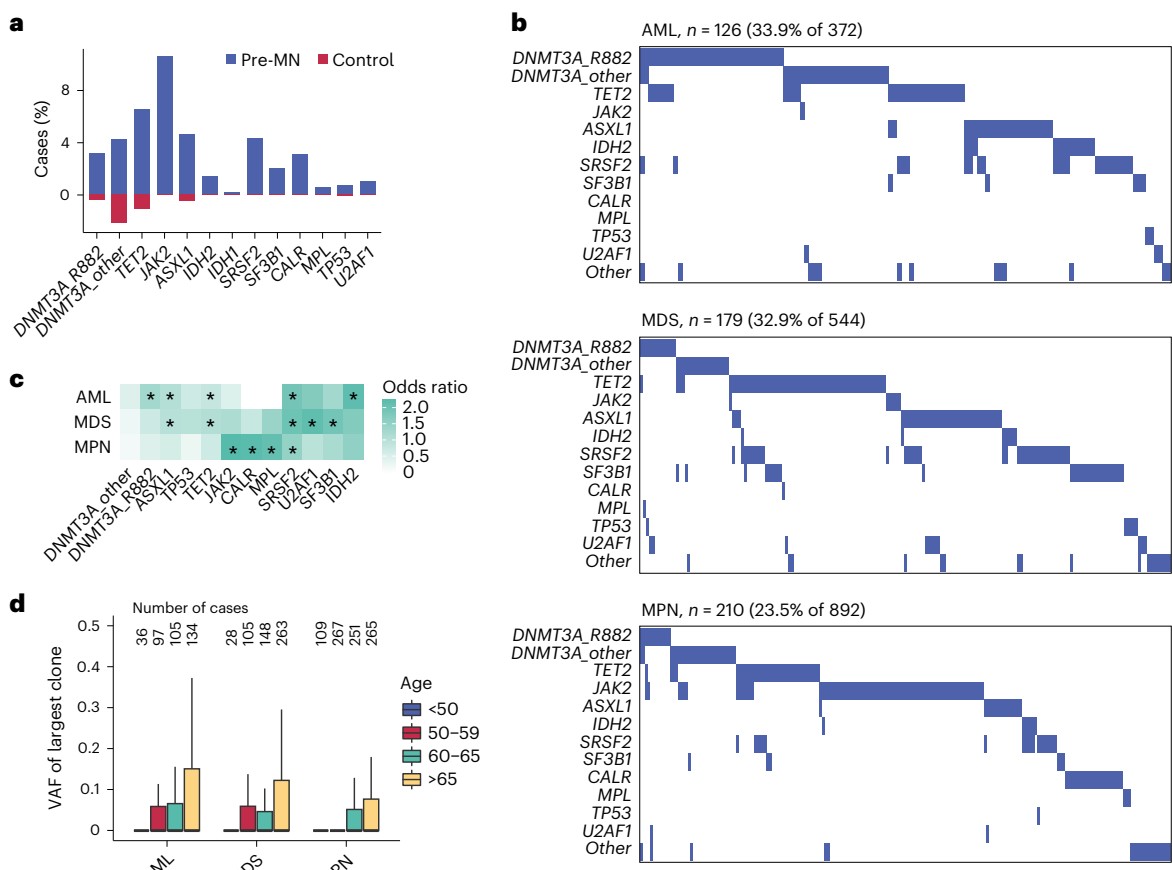

**Fig. 2 | Driver mutations in pre-MN individuals who later developed myeloid neoplasms. a**, Prevalence of common CH driver gene mutations among UKB participants that developed a myeloid neoplasm (pre-MN) compared with controls. **b**, Waterfall plots of mutation profiles in 126 pre-AML, 179 pre-MDS (including pre-CMML) and 210 pre-MPN cases. Each column represents a different pre-MN participant. **c**, Associations between the risk for different types of MN and common driver gene mutations (Fisher's test, *$P < 10^{-10}$; see Supplementary Table 10 for details). **d**, Distribution of clone sizes among different pre-MNs by advancing age. In the box plots, central lines indicate medians, boxes indicate 25–75% quantiles and ranges indicate 1.5 interquartile ranges from the upper or lower quartiles. The numbers of cases in each age bracket are indicated above the box plots.

Among the 1,808 included participants who went on to develop myeloid neoplasm ('pre-MN'), we identified CH mutations in WES from 515 (28.5%), a lower proportion than reported with deep targeted sequencing[8]. By contrast, we identified CH mutations in only 4.8% (21,814 of 451,647) of those who did not develop myeloid neoplasms (controls). In line with previous studies, pre-MN cases commonly had mutations in 'high-risk' genes such as *JAK2*, *SRSF2*, *SF3B1* and *IDH2*, while mutations in controls mainly affected *DNMT3A*, *TET2* and *ASXL1* (Fig. 2a). The proportion of pre-MN participants harboring CH driver mutations was similar among pre-AML (126/372 = 33.9%), pre-MDS (179/544 = 32.9%) and pre-MPN (210/892 = 23.5%) cases. However, there were marked differences in the relative prevalence of different CH driver genes among different types of myeloid neoplasms that reflected their known driver landscapes (Fig. 2b). For example, *DNMT3A* R882 mutations were more common in AML; *TET2*, *SRSF2* and *SF3B1* mutations in MDS and *JAK2*; and *CALR* and *MPL* in MPN (Fig. 2c and Supplementary Fig. 2). Clonal sizes increased with advancing age in all pre-MN subtypes (Fig. 2d).

We previously showed that target gene identity and VAF of driver mutations can be used to predict the risk of developing AML[8]. In addition, we and others found that changes in blood cell counts were also associated with AML risk[4,8], but we were unable to investigate whether combining gene mutations and blood counts can improve prognostication due to limited data availability. Also, the ability to predict the risk of progression to MDS or MPN has not previously been investigated

in this manner. As the UKB captures both gene mutations (genotype) and blood test results (phenotype) from the same individual, we next investigated whether the integration of both data types can improve predictive models of myeloid neoplasm risk. Abbreviations of the parameters are listed in Supplementary Table 5.

Before building myeloid neoplasm risk models, we considered that pre-MDS, pre-AML and pre-MPN cases showed varying or even inverse associations with certain blood count parameters (Supplementary Fig. 3). To account for these divergent associations, we chose to analyze each type of myeloid neoplasm separately. In addition, to streamline onward analyses, we removed highly correlated blood count parameters (Spearman correlation > 0.9), retaining only the parameter most commonly used in clinical reporting (Methods and Supplementary Fig. 4).

We proceeded to quasi-randomly partition the UKB cohort into a training set with 207,035 samples and a validation set with 207,039 samples and then trained time-dependent Cox proportional hazards models on the training set, including death by other causes as a competing risk (Methods). Starting with a core model based solely on age, sex, VAF and mutations in genes previously found to be predictive of progression to myeloid neoplasms[7,8,18], we used forward stepwise regression to iteratively add additional parameters to each of three distinct models for AML, MDS and MPN prediction. Parameters were added to individual models one at a time such that the developing model displayed the highest concordance until the improvement in

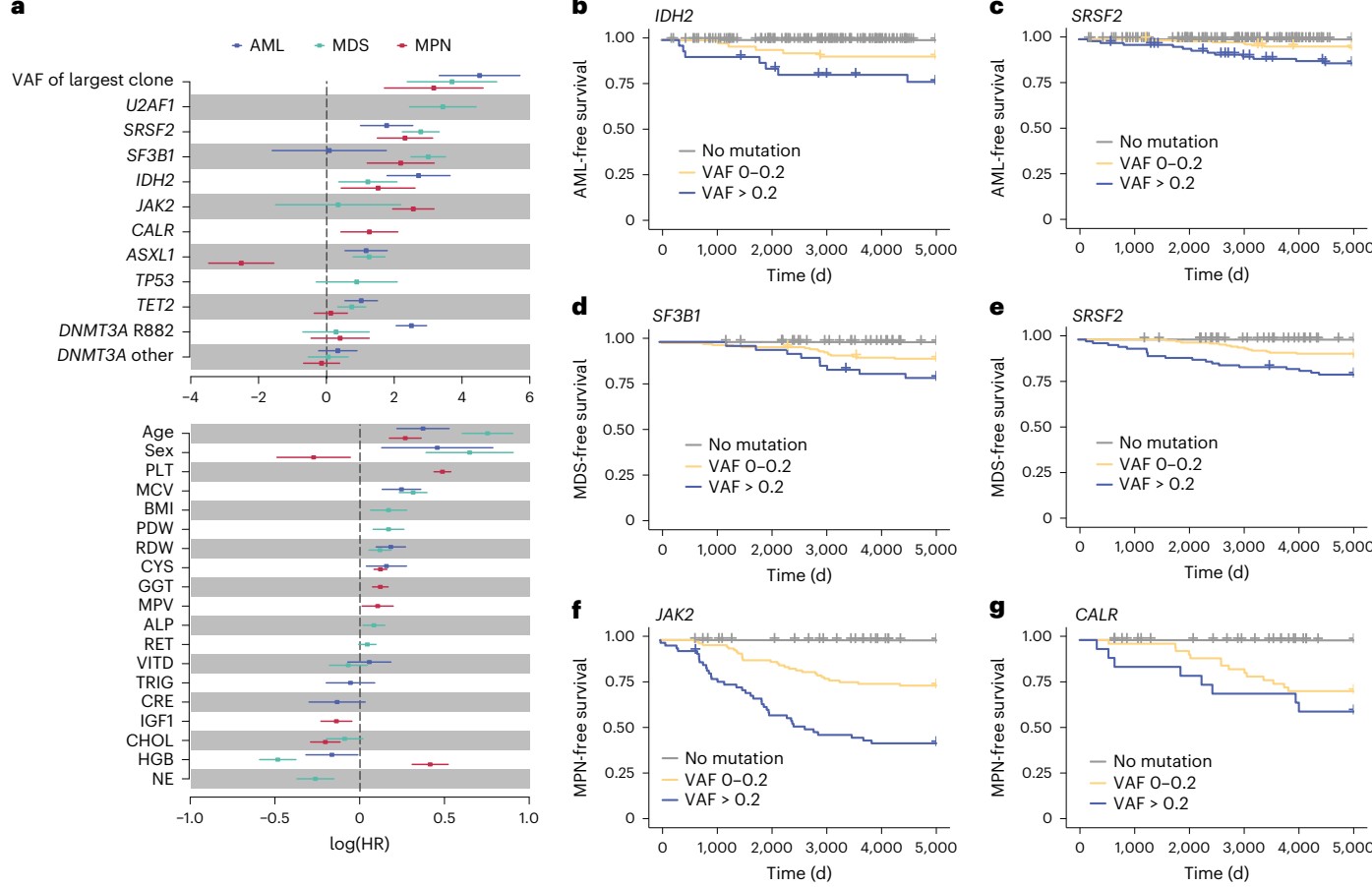

**Fig. 3 | Impact of individual prognostic parameters on myeloid neoplasm prediction. a**, HRs for AML, MDS and MPN, by gene mutation and blood test parameter. The central squares indicate HRs and the lines indicate 5–95% confidence intervals. Only parameters selected by stepwise multivariate regression for inclusion into the relevant model are plotted. **b**–**g**, Kaplan–Meier curves of the most significant genetic predictors by VAF of the driver mutation: *IDH2* and AML-free survival (**b**); *SRSF2* and AML-free survival (**c**); *SF3B1* and MDS-free survival (**d**); *SRSF2* and MDS-free survival (**e**); *JAK2* and MPN-free survival (**f**) and *CALR* and MPN-free survival (**g**). PDW, platelet distribution width; RDW, red cell distribution width; CYS, cystatin-C (serum); GGT, γ-glutamyl transferase (serum); MPV, mean platelet volume (serum); ALP, alkaline phosphatase (serum); VITD, vitamin D (serum); TRIG, triglyceride concentration (serum); CRE, creatinine (serum); IGF1, insulin-like growth factor 1 concentration; NE, neutrophil count.

concordance was less than 0.1% of the total (Methods; Extended Data Fig. 1 and Supplementary Tables 5 and 6).

Using the three final models, we quantified the hazard ratios (HRs) associated with each predictive variable for AML, MDS and MPN. This revealed that HRs associated with individual parameters varied substantially for different myeloid neoplasms (Fig. 3a), something that is also evident when applying univariate analyses (Supplementary Fig. 5). For example, *DNMT3A* R882 mutations were specifically associated with AML, *SF3B1* mutations with MDS and *JAK2/CALR* mutations with MPN (Fig. 3a). By contrast, mutations in genes such as *SRSF2* and *IDH2* afforded similar HRs for different types of myeloid neoplasms. Also, multiple phenotypic features, including increasing age, predicted an increased risk of all myeloid neoplasms. With other parameters such as HGB, higher values predicted an increased MPN risk, while lower values predicted a higher risk of MDS and AML (Fig. 3a). We also found that for many of the higher-risk driver mutations, a higher VAF was associated with a significant decrease in disease-free survival (Fig. 3b–g).

The presence of mosaic chromosomal abnormalities (mCAs) in leukocyte DNA has also been associated with an increased risk of hematological malignancy[19] and we observed significant associations of pre-AML cases with mosaic loss of the long arm of chromosome 5 (−5q), pre-MDS with −5q and 4q loss-of-heterozygosity (4q LOH), and pre-MPN with 9p LOH, +9p and +9 in the UKB (Extended Data

Fig. 2a). Addition of mCAs to our models improved the identification of pre-MNs among individuals with mCAs, while missing a smaller number of pre-MNs who did not have mCA (Extended Data Fig. 2b). However, the addition of mCAs only had a modest effect on overall test performance (Extended Data Fig. 2c–e). In view of this and as mCAs are not routinely captured by standard diagnostic assays, we did not include them in our final models. Furthermore, to test the impact of genetic ancestry on myeloid neoplasm prediction, we incorporated the first five principal components of genetic ancestry into each of the three MN-predictive models and found that this had a negligible effect (Extended Data Fig. 3).

To assess the performance of our models, we run them on the UKB validation set to predict the risk of developing different types of myeloid neoplasms, at any time from recruitment to the end of follow-up (median = 12.6 years). We found that the respective model performed well for predicting future MPN (area under curve (AUC) = 0.82; concordance = 0.81 ± 0.01), MDS (AUC = 0.86; concordance = 0.86 ± 0.01) and AML (AUC = 0.78; concordance = 0.76 ± 0.02; Extended Data Fig. 4a). Similar results were observed using logistic regression models trained in a similar way on the training set, with the exception of AML, for which the Cox regression model performed better (Extended Data Fig. 4b). We also tested random survival forest models trained on all mutational, blood and biochemistry data with three sets of parameters

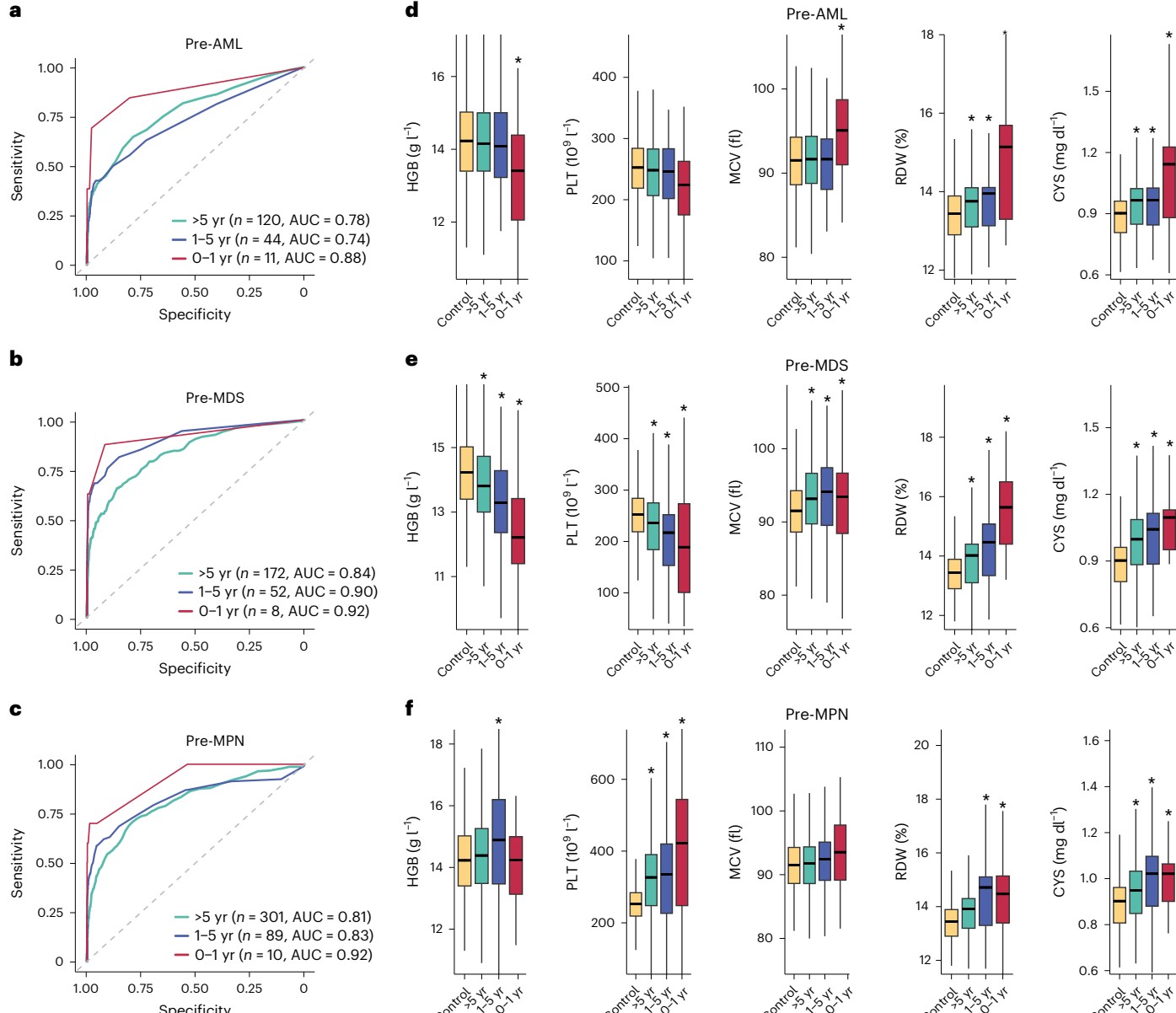

**Fig. 4 | Time-dependency of predictive models and blood parameters in relation to myeloid neoplasm diagnosis. a–c,** Time-dependent ROC curves computed using predicted outcomes on the validation set versus clinical diagnoses of myeloid neoplasm in 0–1 year, 1–5 years and over 5 years after blood sampling in pre-AML (**a**), pre-MDS (**b**) and pre-MPN (**c**) participants. ROC curves were computed using the incident/dynamic method (see Methods for details); $n$ = number of individuals with the relevant diagnosis in the validation set.

**d–f,** Impact of time to diagnosis on the distribution of HGB, PLT, MCV, RDW and CYS in pre-AML (**d**), pre-MDS (**e**) and pre-MPN (**f**) participants, respectively, compared with controls. (*$P < 0.05$ Wilcoxon rank-sum test; see Supplementary Table 10 for details). In the box plots, central lines indicate medians, boxes indicate 25–75% quantiles and ranges indicate 1.5 interquartile ranges from the upper or lower quartiles.

and observed no significant improvement in performance compared with Cox models (Supplementary Fig. 6). Notably, the Cox models performed very similarly on the training and validation sets (Supplementary Fig. 7), indicating there was no significant overfitting or underfitting. Furthermore, the predicted probability of developing a myeloid neoplasm by the end of the follow-up period agreed closely with the observed incidence of myeloid neoplasms in the UKB validation set (Extended Data Fig. 5).

The UKB data are subject to selection biases toward European ancestry, healthy individuals and those who are willing to volunteer, while the measurement of blood, biochemistry and genetic data are subject to batch effects. To validate the performance of our

models outside the UKB, we tested our models on an independent cohort (Leeds CCUS cohort) composed of 204 individuals with CCUS recruited from 2014 to 2016 (138 men and 76 women aged 31–91 years, mean ± s.d. = 74 ± 9.6). Individuals were followed-up until 2019 with a follow-up period of up to 5.5 years (mean ± s.d. = 3.0 ± 1.7), during which 8 individuals developed AML, 35 developed MDS and 1 developed MPN (Supplementary Table 8). We ran our AML and MDS models on this cohort and observed good performance for predicting both AML (AUC = 0.74) and MDS (AUC = 0.73), as well as 'any myeloid neoplasm' (AUC = 0.76; Extended Data Fig. 6a–c). Furthermore, the predicted probability of developing a myeloid neoplasm within 5 years agreed well with the observed fraction of myeloid neoplasm diagnoses in

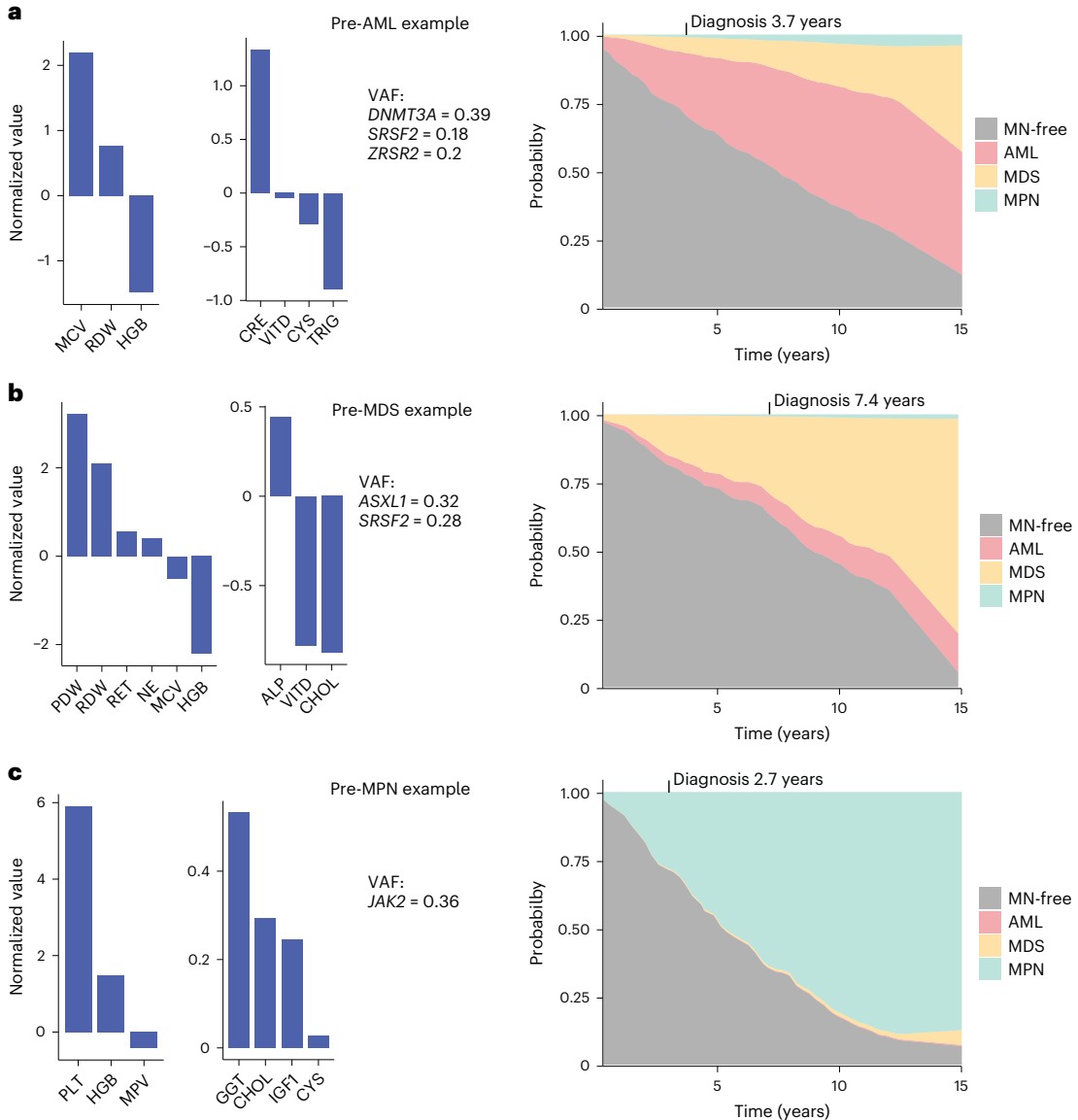

**Fig. 5 | MN-predict, a web-based platform for quantification of future risk of developing myeloid neoplasms. a–c**, Examples of predictions of MN risk by MN-predict in three individuals who went on to develop AML after 3.7 years (**a**), MDS after 7.4 years (**b**) and MPN after 2.7 years (**c**), respectively. The predictions were derived using three separate Cox regression models for predicting AML, MDS and MPN. In each panel, the values of input parameters for the model relevant to the downstream diagnosis are shown on the left (gene mutations, highest VAF and blood tests results depicted as normalized values relative to the median on a log scale) and the actual predictions on the right. The probability of different outcomes is represented by the vertical height of the corresponding color at any given time.

the follow-up period, with a slight over-estimation of 5-year risks of myeloid neoplasms (Extended Data Fig. 6d), which was most likely due to follow-up of most patients being less than 5 years. To overcome this, we then analyzed a separate clinical cohort (Pavia CCUS cohort) containing 312 individuals with CCUS (147 men and 165 women aged 18–89 years, mean ± s.d. = 57 ± 17.3) and a longer follow-up period of up to 15.1 years, during which 49 developed MDS and 2 developed AML (Supplementary Table 9). Our MDS model performed very well in predicting MDS development with a receiver operating characteristics (ROC) curve AUC = 0.84 and a very good agreement between predicted and observed cases of MDS (Extended Data Fig. 7).

Next, to understand the time-dependency of our models, we tested their performance at 1, 2 and 5 years before myeloid neoplasm diagnosis and found that performance generally improved nearer the time of diagnosis, particularly for AML (Fig. 4a–c). To investigate

this further, we looked at how blood test parameters differed by time before diagnosis of a myeloid neoplasm. This revealed that many key blood test parameters changed with time to diagnosis, in patterns that differed between different types of myeloid neoplasms (Fig. 4d–f and Supplementary Figs. 8–10). For example, PLT was substantially higher in pre-MPNs even 10 years before diagnosis, while the corresponding fall in PLT associated with AML was not observed until the final year before diagnosis (Fig. 4d–f). Also, parameters like mean corpuscular volume (MCV) and hemoglobin concentration (HGB) only changed substantially in pre-AML samples during the final year before diagnosis (Fig. 4d), reflecting the improved performance of our AML model during this year. By contrast, for MDS and MPN, many of the predictive parameters were substantially different >5 years before diagnosis.

Finally, to aid clinical hematologists managing patients with high-risk CH, we built a user-friendly web-based application MN-predict

(https://bioinf.stemcells.cam.ac.uk/shiny/vassiliou/MN_predict) that can predict the risk of MN using selected genetic and blood test parameters (Methods). MN-predict enables individualized predictions of the risk of developing different types of myeloid neoplasms over time and also aggregates these predictions to infer the probability of MN-free survival (Fig. 5).

## Discussion

The demonstration that individuals at risk of developing AML can be identified years in advance from the genetic characteristics of their CH clones[7,8] has spurred significant interest in the prospect of myeloid cancer prevention[9,20]. However, less is known about the predictability of myeloid malignancies like MPN and MDS, which also arise from CH[3,4,13], or the prognostic relevance of nongenetic variables such as blood cell counts and biochemical tests/parameters[8].

Here using data from 454,340 UKB participants, we investigate the characteristics of individuals that went on to develop a myeloid neoplasm and use these to construct three separate models for predicting the development of AML, MDS or MPN, which incorporate both genetic and nongenetic variables. We first found that while the CH driver landscape in pre-MN participants reflected that of the onward diagnosis, there was significant overlap among different myeloid neoplasm subtypes. Underlying this, we observed varying strengths of association between particular gene mutations and each of the three myeloid neoplasm subtypes (Fig. 2). For example, *SF3B1* mutations were substantially associated with a higher risk of MDS, while *SRSF2* mutations were substantially associated with all myeloid neoplasm subtypes, with *SRSF2/TET2* comutated cases were more likely to develop MDS and SRSF2/IDH2 comutated cases were more likely to develop AML. Also, *DNMT3A* R882 mutations were specifically associated with AML.

Next, starting with a core model based on age, sex and mutations in CH genes known to be associated with AML risk[8], we used forward stepwise regression to build three Cox regression models for estimating the likelihood of developing AML, MDS and MPN, as well as delineating the risk of different gene mutations in a multivariate context. This revealed that the incorporation of blood test parameters improved model performance. Notably, parameters like HGB had opposite effects on the risk of developing MDS versus MPN, justifying the construction of separate models for these myeloid neoplasm types. Predictive performance (AUC for validation set) of the MDS and MPN models at >1 year and >5 years to diagnosis was better than that of the AML model, while in the final year, all three models performed similarly. In line with this, changes in blood cell counts/indices were evident many years before diagnosis in both pre-MDS and pre-MPN (Fig. 4). In general, the improved model performance nearer the time of myeloid neoplasm diagnosis may reflect the fact that larger clones have a more deterministic behavior than small ones, whose fate is more dependent on chance. A similar conclusion can be drawn from a large study of *JAK2*-V617F mutation frequency, which reported that small clones (VAF ≤ 1%) are a lot more abundant than large ones[21]. Separately, as a further check of model performance, we noted that predicted and observed numbers of myeloid neoplasms in the validation set agreed closely, despite a slightly higher number of MPN diagnoses in the UKB than reported in other European population studies[1,22]. We separately developed and tested predictive models based on logistic regression and random survival forests, which also displayed good predictive performances in our validation set but did not exceed those of the Cox models (Extended Data Fig. 4b and Supplementary Fig. 6).

Next, to ensure that our Cox models perform well in independent datasets, we tested them on two separate clinical CCUS cohorts of 204 (Leeds CCUS cohort) and 312 (Pavia CCUS cohort) patients. Despite having to impute certain missing parameters, we found that our models performed well with both, supporting their generic applicability and suitability for use in real-life clinical cohorts.

Using these Cox models, we then constructed MN-predict, an accessible web-based tool that calculates the likelihood of developing different types of myeloid neoplasms over 15 years after input of age, sex, somatic mutations and a milted set of routine blood test results (Fig. 5). Of note, a contemporaneous study using UKB data developed a different prognostic approach that uses somatic mutations and blood parameters to classify individuals into high, intermediate or low-risk groups for myeloid neoplasms[23]. This is a very valid approach that makes for an easy-to-use clinical tool but provides less granularity compared with MN-Predict as it groups all types of myeloid neoplasms into a single category and does not capture the fact that individuals within the same risk group can have very different likelihoods of progression to myeloid neoplasms. By contrast, MN-Predict can help clinicians to further individualize CH management by providing year-by-year probabilities for each type of myeloid neoplasm over 15 years. Also, by excluding individuals who met diagnostic criteria for MPN diagnosis at UKB entry, MN-Predict gives more realistic estimates of MPN risk.

We anticipate that MN-predict will be of substantial use to researchers and to hematologists managing patients with high-risk CH and CCUS. Users of MN-predict need to be aware that UKB participants display a 'healthy volunteer bias'. However, as epidemiological factors are not major determinants of myeloid neoplasm risk, it is unlikely that prediction accuracy will be substantially affected by this bias. Also, as UKB participants are predominantly of European ancestry (~80%)[24], caution should also be exercised when using MN-predict in other ancestries. The latter is partially mitigated by the fact that the top five principal components of ancestry did not substantially alter model performance.

Collectively, our study represents an important advance in the field of myeloid cancer prediction and provides accessible predictive models that can guide research in this field, assist the management of patients with high-risk CH and help define entry criteria for future interventional studies for myeloid cancer prevention.

## Online content

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

## Methods

### Data acquisition

UKB is a large-scale biomedical database and research resource containing genetic, lifestyle and health information from half a million UK participants. UKB has approval from the North West Multicentre Research Ethics Committee (11/NW/0382) and all participants provided written informed consent. The present study has been conducted under approved UKB application number 56844. Electronic health records were downloaded from the UKB portal in April 2022. For each participant, the disease phenotypes were extracted using the following ICD-9/ICD-10 codes: AML—205.0, 205.2, 205.3, 205.8, 205.9, 206.0, 206.2, 207.0, 207.2, 238.4, 238.5, 238.7, C92.0, C92.2, C92.3, C92.4, C92.5, C92.6, C92.7, C92.8, C92.9, C93.0, C93.2, C94.0 or C94.2; MDS—238.4, 238.5, C94.6 or D46; MPN: 238.7, D45, D47.0, D47.1, D47.3 or D47.4; CMML—206.1, C93.1. The Pavia cohort study was approved by the Ethics Committee of the IRCCS Policlinico San Matteo Foundation, Pavia, Italy (reference: 20180009874). The Leeds cohort study was approved by the North East–York Research Ethics Committee (reference: 16/NE/0105).

### Statistics and reproducibility

Individuals ($n = 129$) with more than one myeloid neoplasm diagnosed within 35 d ($n = 71$, of whom 60 had AML + another myeloid neoplasm), and those diagnosed with AML and then another myeloid neoplasm ($n = 58$, with the second diagnosis made 36–18,39 d later, mean = 497 d), were classed as 'MN-indeterminate' and excluded from analysis, as we wanted to be certain of the specific myeloid neoplasm diagnosis given that our aim was to develop different predictive models for each of the main myeloid neoplasm subtypes. Additionally, 39,465 samples with more than two missing values in blood count and biochemistry data were excluded from modeling to reduce noise.

### Whole-exome sequence data processing, CH mutation calling and filtering

Whole-exome sequencing of blood DNA from 454,340 participants was used to identify somatic mutations using Mutect2 software (GATK version 4.1.3.0) through the DNAnexus platform using Docker image broadinstitute/gatk:4.1.3.0. Mutect2 was run in 'tumor-only' mode with default parameters, over the exon intervals of 38 genes previously associated with CH (Supplementary Table 1). To minimize sequencing artifacts and to filter out potential germline variants, we used a 'panel-of-normals' from the 1000 Genomes Project (1000GP) and the Genome Aggregation Database (gnomAD) obtained from the GATK best practices repository (https://gs://gatk-best-practices/somatic-hg38). Raw variants called by Mutect2 were filtered out with FilterMutectCalls using the estimated prior probability of a reading orientation artifact generated by LearnReadOrientationModel. Putative variants marked 'PASS' by FilterMutectCalls were selected for filtering. Variants marked as 'germline' or 'weak_evidence' were rescued if they were present at least five times in the PASS ones. Gene annotation was performed using Ensembl Variant Effect Predictor (v.102). For identifying CH, we required variants with a minimum number of alternate reads of 2, evidence of the variant on both forward and reverse strands, a minimum depth of 7 reads for SNVs and 10 reads for short indels and substitutions and a minor allele frequency in the population lower than 0.001 (according to 1000GP phase 3 and gnomAD r2.1).

From the resulting calls, we selected those meeting the inclusion criteria established by Vlasschaert et al.[15], to optimize the exclusion of germline variants and sequencing artifacts (Supplementary Table 1). For *TET2* and *CBL*, for which individual driver definitions are not exhaustively defined, variants were removed if they failed a one-sided exact binomial test ($P > 0.01$), where the null hypothesis was that the number of alternative reads supporting the mutation were 50% of the total number of reads. Variants with $n \leq 20$ were all retained. To find the best cut-off for the minimum number of reads required to call a CH

mutation, we tested three different cut-offs: ≥2, ≥3 and ≥5 reads on Mutect2 output and found that ≥2 read is most suitable for our study as it improved concordance indices of our AML model while leaving the MDS and MPN model performance unchanged (Supplementary Fig. 11).

Samtools mpileup (version 1.15) was used to capture single-nucleotide variations (SNVs) at 22 known hotspots (Supplementary Table 2), including *U2AF1* SNVs that were missed due to a mistake in the human GRCh38 assembly[16]. SNVs with ≥3 reads and VAF > 0.1 were retained and used for predictive models. Additionally, 4-nucleotide-insertions in NPM1 within the range of chr5:171410538-171410544 were examined manually with prior knowledge of the common 4-nt inserts and only two known cases were identified[25].

mCA data were obtained from the UKB Application 19808 Return 3094 (ref. 26). Associations between mCAs and myeloid neoplasms were tested using Fisher's exact test. Significant mCAs ($P < 0.00001$) were extracted, including chromosome 1p LOH, 4q LOH, 5q loss, 7q LOH, 8 gain, 9p LOH, 9 gain, 12q loss, 14q LOH, 17q loss and 20q loss. X- and Y-chromosome mCAs were not investigated.

### Predictive modeling for myeloid neoplasms

All data types used in model development with explanations of relevant abbreviations are provided in Supplementary Table 5.

To optimize model performance, 39,465 samples with more than 2 missing values in blood count and biochemistry data ($n = 39,283$ controls and $n = 171$ pre-MNs) were excluded from modeling. Next, we removed interderivable variables, namely RBC, MCH and HT, from the complete blood count results and retained HGB, MCV and MCHC. Missing values were imputed using the median of the UKB population. We excluded individuals who had a myeloid neoplasm diagnosis before blood collection ($n = 648$), individuals whose blood test results were consistent with a probable diagnosis of polycythemia vera ($n = 26$; HGB > 16.5 g dl$^{-1}$ and with *JAK2*-V617F) or essential thrombocythemia ($n = 82$; PLT > $450 \times 10^9$ l$^{-1}$ and with *JAK2*-V617F/*CALR*/*MPL* mutations) and individuals ($n = 129$) with more than one myeloid neoplasm diagnosed within 35 d or with AML and then another myeloid neoplasm. While it is possible that some additional UKB participants with slightly abnormal blood counts at study entry had a myeloid neoplasm (for example, MDS), we had no way to identify them and also note that their blood test results did not trigger a clinical referral. Samples of remaining individuals were used to test for linear dependency between each pair of parameters of phenotypic variables within the entire dataset and within each type of myeloid neoplasm using Spearman correlation (Supplementary Fig. 4). For each highly dependent pair or cluster (Spearman correlation > 0.9 in all myeloid neoplasms), we selected the most commonly used parameter in clinics and excluded the others, retaining PLT over plateletcrit (PCT), reticulocyte count (RET) over high light scatter reticulocytes (HLR) and cholesterol (CHOL (serum)) over apolipoprotein B/low-density lipoprotein direct. We did not attempt to distinguish between CH and CCUS in our models, as blood test results that define CCUS are included and as a formal CCUS diagnosis requires persistence of cytopenia over several months as well as the clinical exclusion of other etiologies[17,27].

Samples were split quasi-randomly into training and validation sets to obtain roughly equal numbers of cases of pre-AML, pre-MDS, pre-MPN and pre-CMML in each set. Specifically, we first separated each type of pre-MN and controls, and then randomly split each into two similar size sets using the random function (Math.random() in Java). We then merged one control with one pre-MN set to generate the training set of 207,035 samples and a validation set of 207,039 samples. All subsequent model development was performed on the training set using both genotype and phenotype parameters and model performance was tested on the validation set. For each type of myeloid neoplasm, an initial Cox proportional hazards model was trained using the R package of 'survival' with all 38 binary genotypic variables (Supplementary Table 1), 30 continuous preselected phenotypic variables

(Supplementary Table 5), sex, age, body mass index (BMI) and the highest VAF. All continuous variables including phenotype, age and BMI were standardized using the following:

$$x'_i = \frac{x_i - \mathrm{Med}(x)}{\sigma(x)}$$

where Med(x) is the median and $\sigma(x)$ is the standard deviation of the variable. A Cox proportional cause-specific hazard model was used for each of the myeloid malignancies, considering death by other causes before the end of the censoring period as a competing risk. To reduce the number of variables in the final model, we used forward stepwise regressions starting with a set of 13 MN-related variables, namely sex, age, VAF and somatic mutations in any of 11 genes that were commonly mutated in CH and/or known to be associated with progression to myeloid neoplasms[7,8,18] (*DNMT3A, JAK2, MPL, CALR, SRSF2, SF3B1, IDH2, TP53, TET2, ASXL1* and *U2AF1*). To avoid overfitting, we excluded genes with ≤4 mutations (that is, ≤2 mutations in the training or validation set) in the relevant pre-MN sample group, namely *JAK2, MPL, CALR* and *U2AF1* from pre-AML, *CALR* and *MPL* from pre-MDS, and *TP53, MPL* and *U2AF1* from pre-MPN. Then from the candidate pool of the 27 remaining genes, BMI and 30 blood/biochemistry parameters, we proceeded to iteratively add one variable to the model at a time. In each iteration, we added each of the *n* variables to the starting set, resulting in *n* sets of variables and trained *n* Cox models on these sets. Of the *n* models, we selected the one with the highest concordance index (C-index)[28] as the new starting set and removed the newly added variable from the candidate pool for the next iteration. We drew the threshold at the iteration where the increase in C-index was <0.1% of the maximum increase of all iterations (that is, the highest C-index of all iterations minus the C-index of the starting concordance), with the variables in that iteration chosen for the final model (Supplementary Table 6). To test the performance of the final models, we constructed time-dependent ROC curves by examining three groups of individuals who developed myeloid neoplasms 0–1 year, 1–5 years and >5 years after the blood assessment separately. For each group, 'observed positives' were defined as the individuals who developed this myeloid neoplasm within this period and 'observed negatives' were defined as the ones who developed this myeloid neoplasm outside this period or ones who never developed the myeloid neoplasm. Predicted probabilities of developing myeloid neoplasms in a time period were calculated as the average of predicted values of all time points within this period from the outcome of Cox regression models. By varying the threshold of predicted probability from its lowest to highest, we compared predicted positives/negatives with observed positives/negatives to obtain pairs of sensitivity and specificity and plotted the ROC curves.

Additionally, we used logistic regressions with the 'glm' function of R to obtain similar results as Cox proportional hazard models. We also trained models with random survival forest on the training set using the 'randomForestSRC' package of R. We scanned three sets of parameters across various numbers of trees (that is *n*(tree)), and numbers of node splits per tree (that is *n*(split)) for each model: *n*(tree) = 50 and *n*(split) = 10; *n*(tree) = 100 and *n*(split) = 10; *n*(tree) = 100, *n*(split) = 20. Time-dependent ROC curves were constructed using the same method as described.

### Validation on independent cohorts
To validate our models, we obtained the genotype, blood and biochemistry data of 204 individuals with CCUS, including 7 pre-AML, 31 pre-MDS and 1 pre-MPN cases (Leeds CCUS cohort). Available genotypic parameters were mutations in genes *DNMT3A, IDH2, TET2, U2AF1, ASXL1, SRSF2, JAK2, TP53, SF3B1, CALR* and *MPL* and VAFs of the largest clone. Available phenotypic parameters are sex, age, MCV, PLT and HGB. Missing phenotypic parameters were imputed as the median of the UKB population and input parameters were processed in the same way as we processed UKB data. We applied all three models to this cohort and compared the predicted probabilities of developing each

type of myeloid neoplasm in the next 5 years with observed myeloid neoplasm diagnosis in the follow-up period (up to 5.5 years), using the same methods as we used for the UKB analysis.

To validate the MDS model, we obtained an independent cohort of 312 individuals, containing 49 cases of pre-MDS and 263 control cases (Pavia CCUS cohort). Available genotypic parameters were mutations in genes DNMT3A, *SRSF2, SF3B1, IDH2, TP53, TET2, ASXL1, U2AF1, JAK2, MPL* and *CALR* and VAFs of the largest clone. Available phenotypic parameters include age, sex, PLT, HGB, MCV and NE. Missing phenotypic values were imputed as the median of the UKB population and input parameters were processed in the same way as we processed UKB data. To validate our MDS model, we applied the MDS model to this cohort and compared the predicted probabilities of developing each type of myeloid neoplasm in the next 15 years with observed myeloid neoplasm diagnosis in the follow-up period (up to 15.1 years), using the same methods as we used for the UKB analysis.

### MN-predict: a web-based myeloid neoplasm risk calculator
As CH can progress to any of the main types of myeloid neoplasms, it would be useful to assess the probability of progression to any of the myeloid neoplasm subtypes for each individual with CH. To achieve this and to provide a one-stop predictive tool for researchers and clinicians managing high-risk CH, we built MN-predict, an accessible web-based tool that generates time-dependent predictions of future risk of progression to AML, MDS or MPN. To do this, we amalgamated the probabilities of developing each of the three myeloid neoplasm subtypes calculated from their respective models using the following approach:

Disease-free survival probabilities for each myeloid neoplasm are predicted as a function of time and the overall probability of getting myeloid neoplasm *x* (where *x* is AML, MDS or MPN) at time point *t* is calculated as

$$\mathrm{Pr}(x,t) = \left(1 - \prod_{i \in (\mathrm{AML,MDS,MPN})} \mathrm{Surv}(i,t)\right) \frac{1 - \mathrm{Surv}(x,t)}{\sum_{i \in (\mathrm{AML,MDS,MPN})}(1 - \mathrm{Surv}(i,t))}$$

where Surv(*x*,*t*) is the probability of disease-free survival for each of the myeloid neoplasm subtypes at time point *t*.

After inputting the genotypic and phenotypic parameters included in their respective Cox models, the MN-predict website generates time-dependent plots of projected probabilities for developing AML, MDS and MPN (or remaining MN-free) over 15 years.

### Reporting summary
Further information on research design is available in the Nature Portfolio Reporting Summary linked to this article.

### Data availability
Individual-level UK Biobank data can be requested via application to the UK Biobank (https://www.ukbiobank.ac.uk). The CH call has been returned to the UK Biobank to enable individual-level data linkage for approved UK Biobank applications.

### Code availability
The MN-predict web application is hosted at https://bioinf.stemcells.cam.ac.uk/shiny/vassiliou/MN_predict. Codes for analyses and figure reproduction have been uploaded to https://github.com/muxingu/mnpredict_paper.

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

## Acknowledgements

This work was funded by an Early Detection Project Grant from Cancer Research UK (EDDCPJT\100010) and a joint grant from the Leukemia and Lymphoma Society (RTF6006-19), and the Rising Tide Foundation for Clinical Cancer Research (CCR-18-500) awarded to G.S.V. The Cambridge Stem Cell Institute is supported by the Wellcome Trust (203151/Z/16/Z, 203151/A/16/Z) and the UKRI Medical Research Council (MC_PC_17230). W.G.D is funded by a Clinical Research Fellowship from the Cancer Research UK Cambridge Centre (CTRQQR-2021\100012). S.P.K. is supported by a UK Research and Innovation (UKRI) Future Leaders Fellowship (MR/T043202/1). L.M was supported by the Associazione Italiana per la Ricerca sul Cancro (AIRC), (grant 20125; AIRC 5x1000 project 21267); Cancer Research UK, FC AECC and AIRC under the International Accelerator Award Program (C355/A26819 and 22796). P.M.Q. is funded by the Miguel Servet Program (CP20/00130). A.S. is funded by Cancer Research UK (grant 29685) and Blood Cancer UK (grant 503). G.S.V. is supported by a Cancer Research UK Senior Cancer Fellowship (C22324/A23015) and work in his laboratory is also funded by the European Research Council, Kay Kendall Leukemia Fund, Blood Cancer UK and the Wellcome Trust. This research was conducted using the UK Biobank resource under approved application 56844. We thank the participants and investigators involved in the UK Biobank resource and in the other genome-wide association studies cited in this work who collectively made this research possible.

## Author contributions

G.S.V. conceived, designed and supervised the study, with help from P.M.Q., M. Gu carried out data analyses and mutation calling, developed and tested regression models, built the MN-predict website, and generated tables and figures. S.C. performed targeted hotspot mutation analysis and extracted phenotype data from UKB. P.M.Q. managed UKB data access, wrote code for mutation calling, carried out mutation filtering and helped with regression models. M. Gerstung gave expert advice on predictive/regression model training and validation. S.V., W.G.D., L. Marando, C.B., I.M., S.P.K., M.A.F. and M. Gerstung contributed ideas and analytical advice throughout the project. M. Gu, P.M.Q. and G.S.V. wrote the paper with help from all co-authors. A.S., C.A.C. and L. Malcovati provided independent cohorts for validation. All authors approved the final version of the paper.

## Competing interests

G.S.V. is a consultant to STRM.BIO and holds a research grant from AstraZeneca for research unrelated to that presented here. M.A.F. is an employee and stockholder of AstraZeneca. The other authors declare no competing interests.

## Additional information

**Extended data** is available for this paper at https://doi.org/10.1038/s41588-023-01472-1.

**Correspondence and requests for materials** should be addressed to Pedro M. Quiros or George S. Vassiliou.

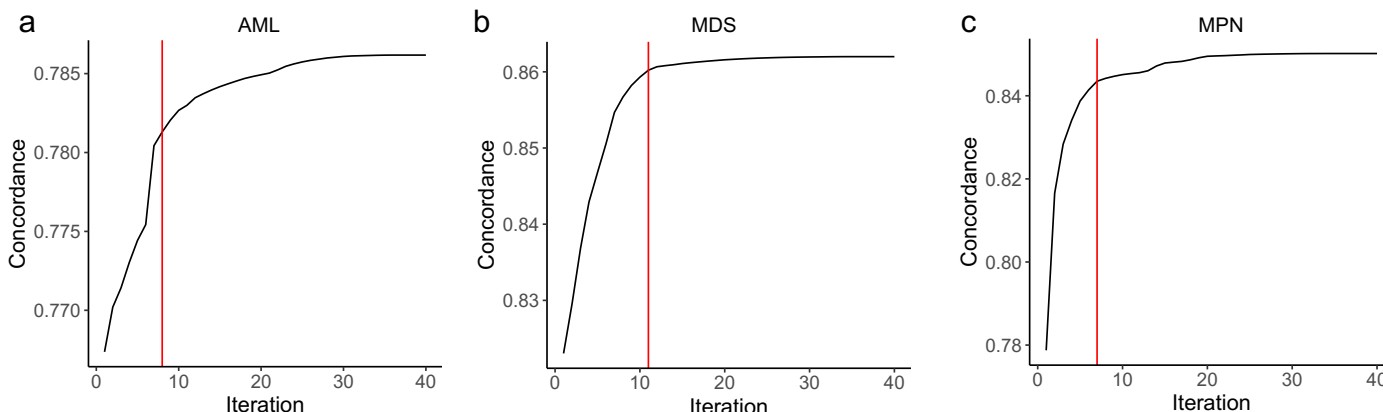

**Extended Data Fig. 1 | Feature selection in different pre-MN models using stepwise regression.** Improvement in concordance by the stepwise addition of predictive variables to the core Cox regression model for developing disease-specific Cox regression models for: **(a)** AML, **(b)** MDS and **(c)** MPN. Variables were added one at a time, such that each iteration resulted in the greatest improvement in concordance index until the increase in concordance <0.1% of the maximum increase of all iterations. The iterations (that is number of additional variables) used in the final models are indicated by the red lines.

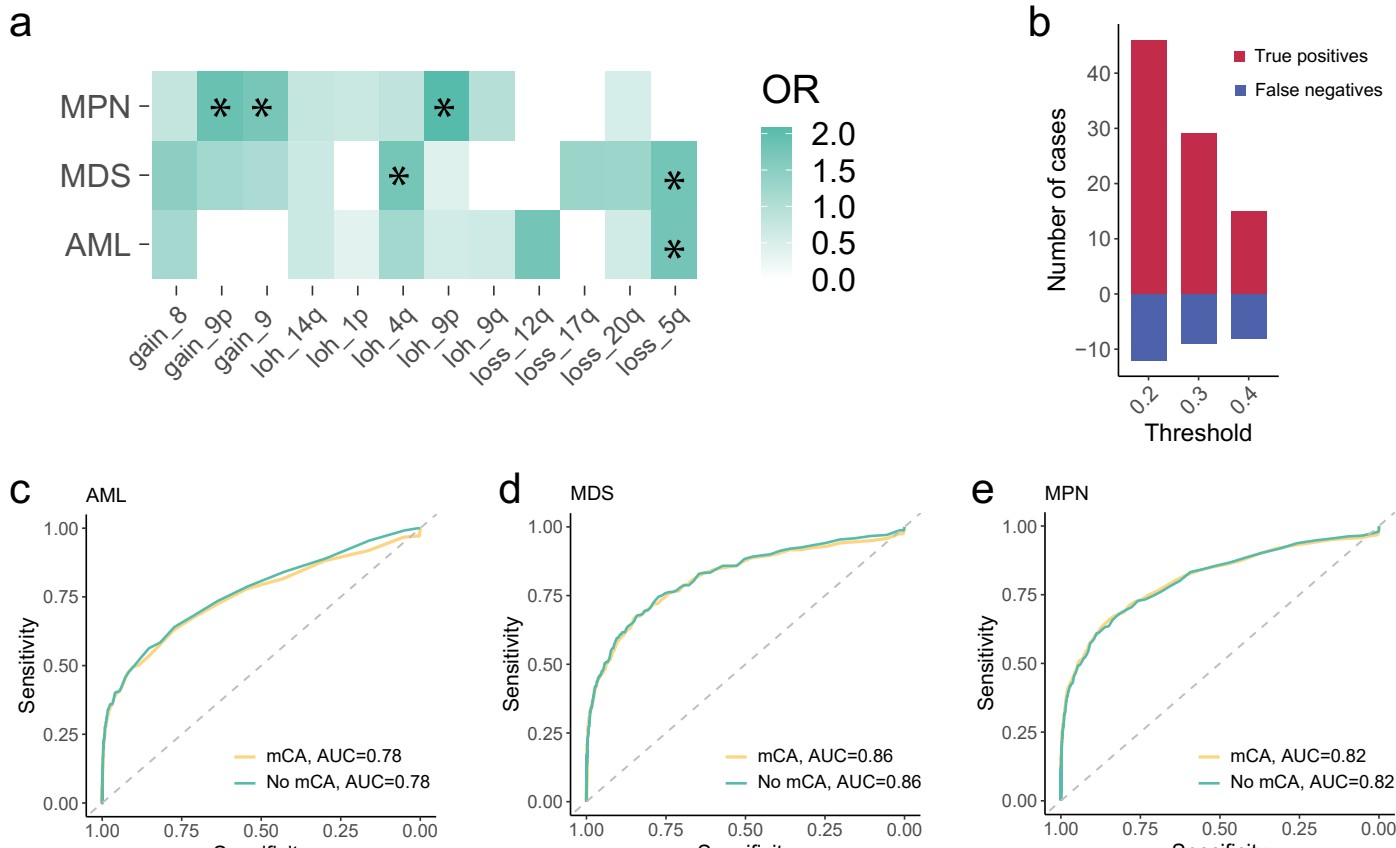

**Extended Data Fig. 2 | Impact of mosaic chromosomal abnormalities on MN prediction models. (a)** Associations between the risk for different types of MN and mosaic chromosomal alterations (mCA, * = Fisher's test p < 10[-5], see Supplementary Table 10 for details; OR = odds ratio). **(b)** Number of true pre-MN cases whose prediction changed by the inclusion of mCAs to the models. We calculated differences between 15-year MN-free survival probabilities of models including mCAs (with mCA) vs excluding mCAs (without mCA). We then tested three thresholds for the difference in MN probability between the two models.

The lowest probability difference of 0.2 led to the correct identification of an additional ~45 pre-MN cases (true positives), at the expense of missing 12 such cases (false negatives). Higher difference thresholds still identified more true positives than false negatives. **(c–e)** Inclusion of mCA to our MN prediction models did not significantly improve model performance as assessed by area under curve (AUC) of recover operating curve for **(c)** AML, **(d)** MDS or **(e)** MPN. Dotted diagonal lines indicate AUC = 0.5.

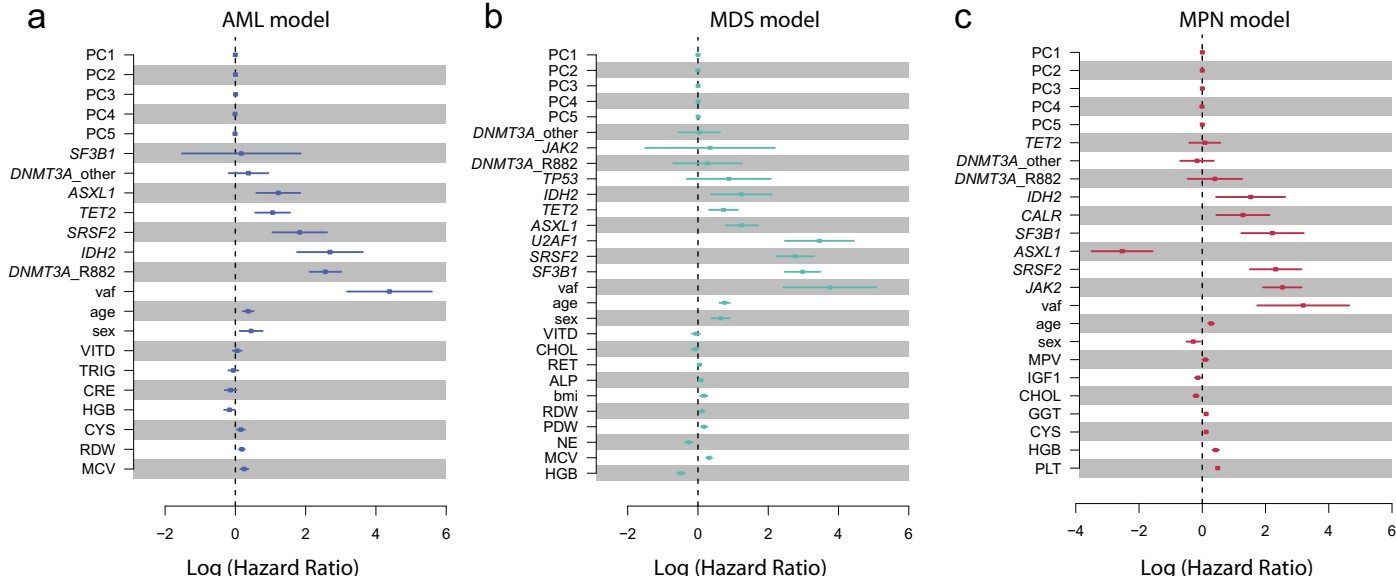

**Extended Data Fig. 3 | Genetic ancestry does not have a major impact on MN prediction models.** Hazard ratios (HRs) associated with predictive variables, after incorporation of the first five principal components of genetic ancestry (PC1-PC5) into MN predictive models for: **(a)** AML, **(b)** MDS and **(c)** MPN. The plots show that ancestry has a negligible impact on these models, with HRs close to 1 (Log1 = 0). Central squares indicate estimated HRs and lines represent the 5–95% confidence intervals. VAF = variant allele frequency of the largest clone. The central squares indicate hazard ratios and the lines indicate 5–95% confidence intervals. Vertical dotted lines indicate HR = 1. Abbreviations for blood/biochemistry parameters are defined in Supplementary Table 5.

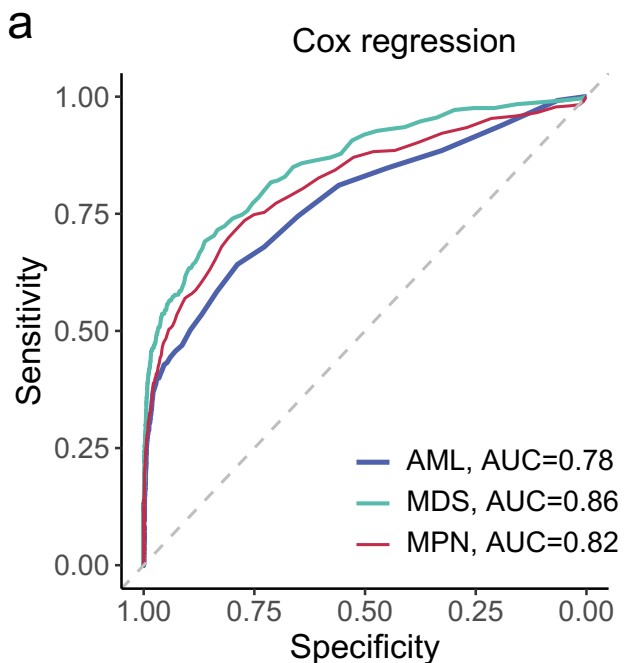

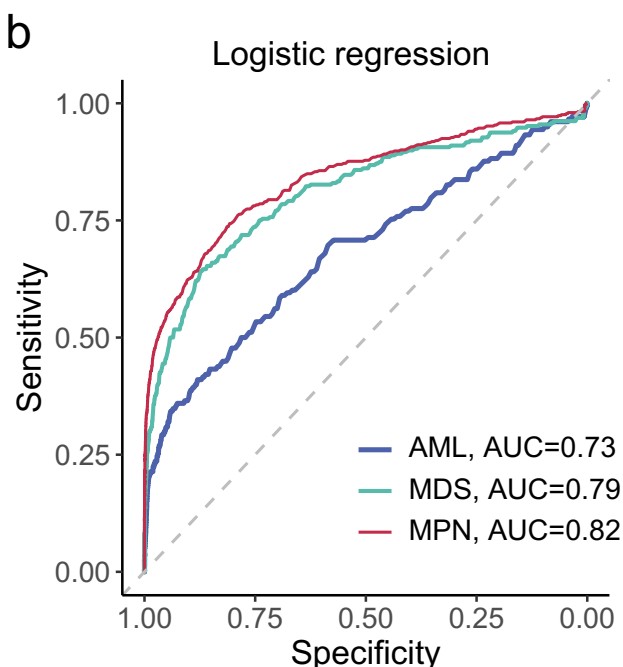

**Extended Data Fig. 4 | Comparison of Cox to logistic regression models for MN prediction. (a)** Recover operating curve (ROC) curves from Cox proportional hazard models for prediction of progression to AML, MDS and MPN. **(b)** ROC curves from logistic regression models. To make the models comparable, we used MN outcomes at any time to the end of the study to compute ROC curves. AUC = area under curve. Dotted diagonal lines indicate AUC = 0.5.

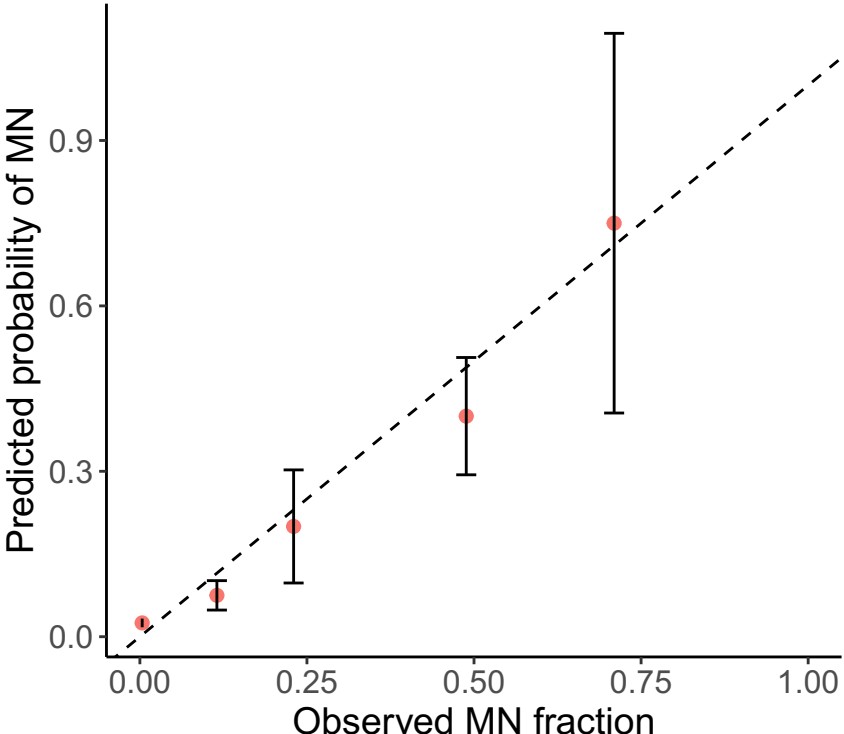

**Extended Data Fig. 5 | Close agreement between prediction and actual incidence of MN.** Comparison of the predicted probability of developing any MN with the observed MN incidence in the UKB validation cohort of 207,039 individuals at any time during the follow-up/observation period (dots showing the mean and error bars showing 1.96 standard deviations that is 5–95% CI). Samples were binned according to predicted probability ranges as follows: 0–0.05, 0.05–0.1, 0.1–0.3, 0.3–0.5 and 0.5–1. Individuals who died during the observation period without having developed MN were not included in the calculations. The plot shows close agreement (along the dotted line y = x) between prediction and observed incidence.

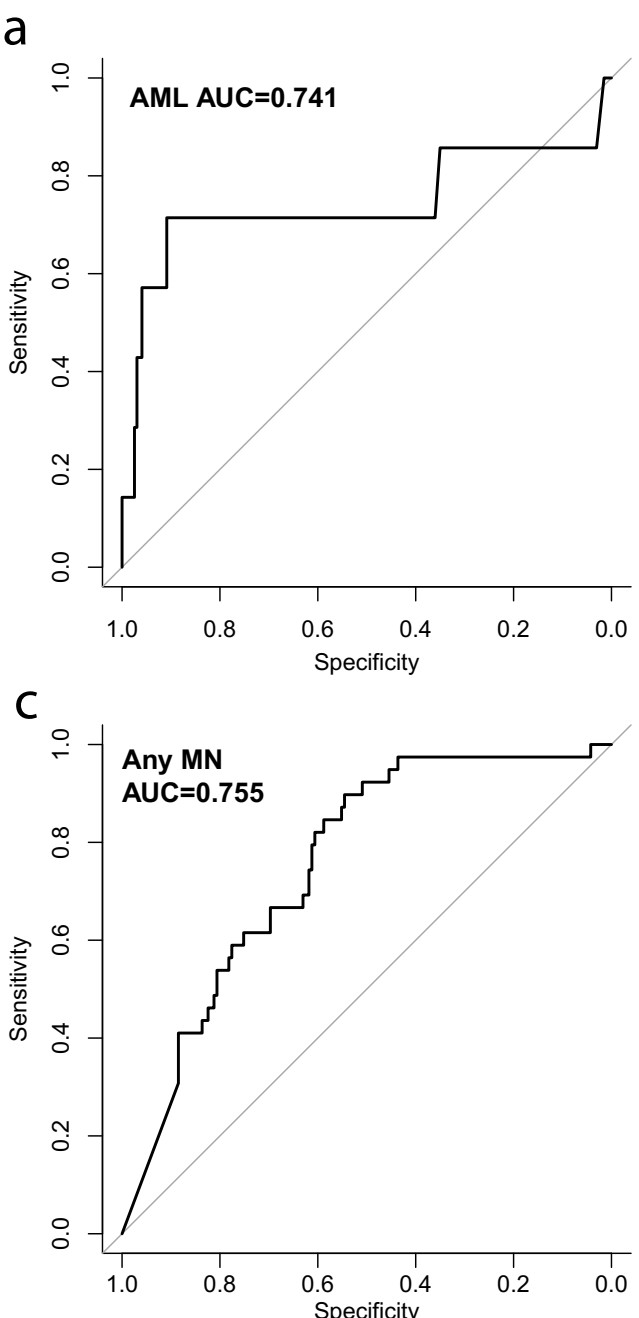

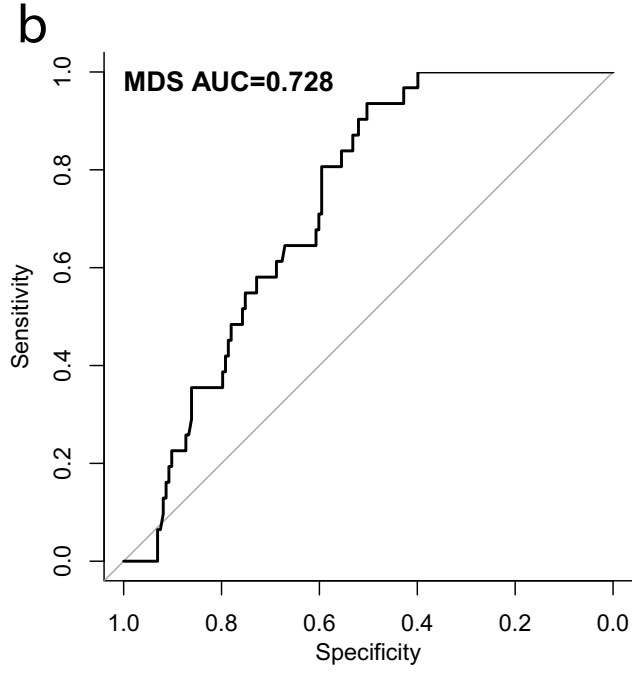

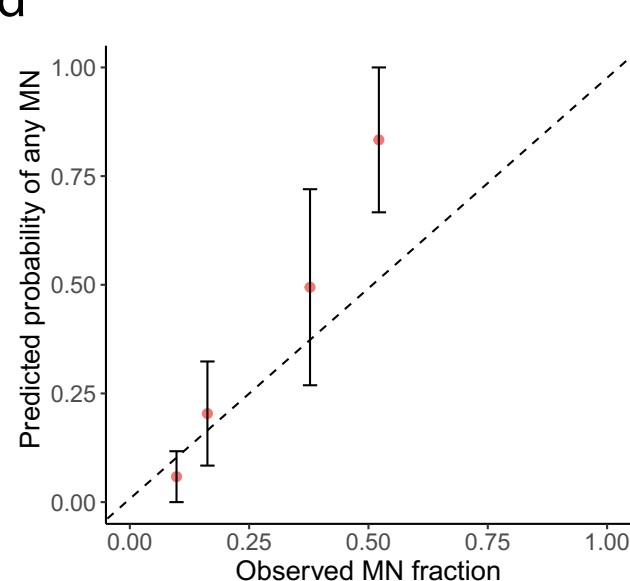

**Extended Data Fig. 6 | Validation of models on the Leeds CCUS cohort.**
**(a-c)** Receiver Operating Characteristics (ROC) curves of the independent cohort computed from predicted probabilities in 5 years versus clinical diagnosis of individuals who developed MN within 5 years after blood sampling. AUC=area under curve. **(a)** AML model. **(b)** MDS model. **(c)** ROC curves of combined probabilities of any MN versus clinical diagnosis. Diagonal lines indicate

AUC = 0.5. **(d)** Comparison of the predicted probability of developing any MN in the next 5 years with the observed MN diagnosed at any time during the follow-up period (dots showing the mean and error bars showing 1.96 standard deviations that is 5–95% CI). Individuals who died before the end of the follow-up period without developing any MN were excluded from the calculation.

a

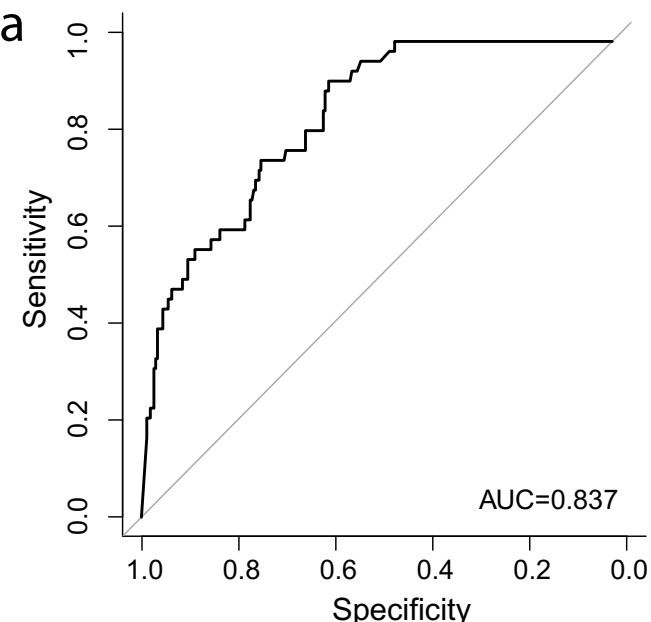

b

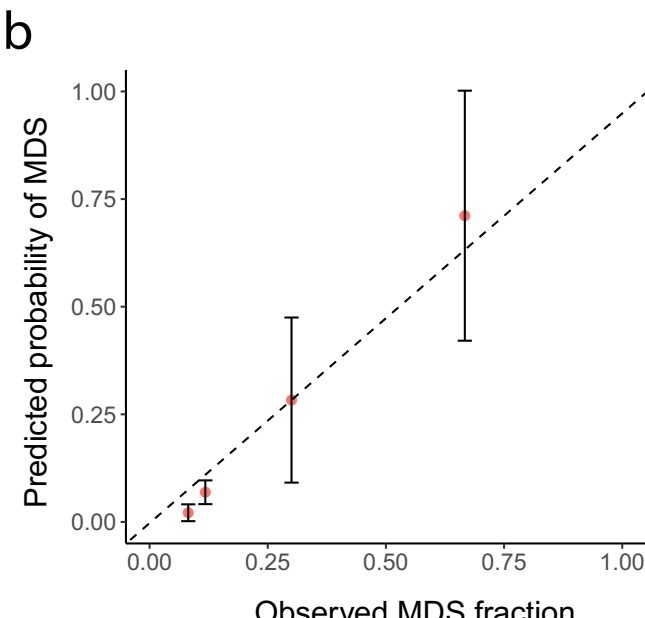

**Extended Data Fig. 7 | Validation of MDS model on the Pavia CCUS cohort.**
**(a)** ROC curve of Cox proportional hazard model for MDS prediction established
from predicted 15-year probability of developing MDS and diagnosis by the end
of the 15-year follow-up period. AUC = area under curve. Diagonal line indicates
AUC = 0.5. **(b)** Comparison of the predicted MDS probability and observed MDS

incident at any time during the follow-up period (dots showing the mean and
error bars showing 1.96 standard deviations that is 5–95% CI). Individuals who
died before the end of the follow-up period without developing any MDS were
excluded from the calculation. Dotted line shows y = x.

# Reporting Summary

## Statistics

For all statistical analyses, confirm that the following items are present in the figure legend, table legend, main text, or Methods section.

| n/a | Confirmed | |
|---|---|---|
| ☐ | ☒ | The exact sample size (*n*) for each experimental group/condition, given as a discrete number and unit of measurement |
| ☒ | ☐ | A statement on whether measurements were taken from distinct samples or whether the same sample was measured repeatedly |
| ☐ | ☒ | The statistical test(s) used AND whether they are one- or two-sided *Only common tests should be described solely by name; describe more complex techniques in the Methods section.* |
| ☐ | ☒ | A description of all covariates tested |
| ☒ | ☐ | A description of any assumptions or corrections, such as tests of normality and adjustment for multiple comparisons |
| ☐ | ☒ | A full description of the statistical parameters including central tendency (e.g. means) or other basic estimates (e.g. regression coefficient) AND variation (e.g. standard deviation) or associated estimates of uncertainty (e.g. confidence intervals) |
| ☐ | ☒ | For null hypothesis testing, the test statistic (e.g. *F*, *t*, *r*) with confidence intervals, effect sizes, degrees of freedom and *P* value noted *Give P values as exact values whenever suitable.* |
| ☒ | ☐ | For Bayesian analysis, information on the choice of priors and Markov chain Monte Carlo settings |
| ☒ | ☐ | For hierarchical and complex designs, identification of the appropriate level for tests and full reporting of outcomes |
| ☐ | ☒ | Estimates of effect sizes (e.g. Cohen's *d*, Pearson's *r*), indicating how they were calculated |

*Our web collection on statistics for biologists contains articles on many of the points above.*

## Software and code

Policy information about availability of computer code

| Data collection | No raw data was downloaded. Analysis was performed using UKB data accessed via DNA Nexus (ukbiobank.dnanexus.com) |
|---|---|
| Data analysis | GATK v4.1.3.0, Mutect2. Samtools 1.15. Custom code for training Cox regression models can be found here: github.com/muxingu/mnpredict_paper. |

For manuscripts utilizing custom algorithms or software that are central to the research but not yet described in published literature, software must be made available to editors and reviewers. We strongly encourage code deposition in a community repository (e.g. GitHub). See the Nature Portfolio guidelines for submitting code & software for further information.

## Data

Policy information about availability of data

All manuscripts must include a data availability statement. This statement should provide the following information, where applicable:
- Accession codes, unique identifiers, or web links for publicly available datasets
- A description of any restrictions on data availability
- For clinical datasets or third party data, please ensure that the statement adheres to our policy

*Provide your data availability statement here.*

# Research involving human participants, their data, or biological material

Policy information about studies with human participants or human data. See also policy information about sex, gender (identity/presentation), and sexual orientation and race, ethnicity and racism.

| | |
|---|---|
| Reporting on sex and gender | We used sex as a variable in our regression/predictive models as male sex is associated with an increased risk of myeloid neoplasia. Individual level information on sex is available from the UKB. |
| Reporting on race, ethnicity, or other socially relevant groupings | The UK Biobank is primarily populated by individuals of European ancestry (~85%). |
| Population characteristics | The UK Biobank is a prospective longitudinal study containing in-depth genetic and health information from ~half a million UK participants. For this study, we analysed 454,340 individuals who had whole-exome sequencing (WES) data released as of March 2022 (age range: 38-72, mean age: 56.5; 54.1% female; ~83% White British). |
| Recruitment | As stated above, the UK Biobank is a prospective longitudinal study containing in-depth genetic and health information from ~500,000 UK participants. Details of UK Biobank participant recruitment are available at: https://www.ukbiobank.ac.uk and from Sudlow C, et al. (2015) PLoS Med 12(3): e1001779. For this study, we analyzed 454,340 individuals who had whole-exome sequencing (WES) data released as of March 2022. Notably, participants were not selected in any way, however as is the case for several such cohorts, there is evidence of selection bias in favor of healthier, older, female, socio-economically better off volunteers (Fry A, et al. Am. J. of Epidemiol., Vol. 186, Issue 9, 1 Nov. 2017, Pgs. 1026–1034). Also, despite a relatively low response rate to invitations to participate, it has been shown that risk factor associations identified in the UK Biobank are generalizable (Batty GD, et al. BMJ 2020; 368:m131). |
| Ethics oversight | The UK Biobank study has been approved by the North West Multicentre Research Ethics Committee (11/NW/0382). All participants provided written informed consent to the UK Biobank. Further details can be found at: https://www.ukbiobank.ac.uk/learn-more-about-uk-biobank/about-us/ethics. The current study has been conducted under approved UK Biobank application numbers 56844. |

Note that full information on the approval of the study protocol must also be provided in the manuscript.

# Field-specific reporting

Please select the one below that is the best fit for your research. If you are not sure, read the appropriate sections before making your selection.

☒ Life sciences      ☐ Behavioural & social sciences      ☐ Ecological, evolutionary & environmental sciences

For a reference copy of the document with all sections, see nature.com/documents/nr-reporting-summary-flat.pdf

# Life sciences study design

All studies must disclose on these points even when the disclosure is negative.

| | |
|---|---|
| Sample size | For this study, we analyzed 454,340 individuals who had whole-exome sequencing (WES) data released as of March 2022. |
| Data exclusions | Participants that developed myeloid neoplasia prior to recruitment were excluded from model training & validation. Also, individuals with more than two missing values in their blood cell count or biochemistry datasets were excluded. We also removed 108 UKB participants with blood test results consistent with a diagnosis of MPN at the time of recruitment from model training and validation. |
| Replication | We divided the 414,074 eligible (non-excluded) participants into a training set of 207,035 and a validation set of 207,039. Models were trained on the training set and validated on the validation set. |
| Randomization | The division of participants into training and validation sets was random and performed using the Math.random() function from Java. |
| Blinding | Not applicable, as the analyses presented in this manuscript do not include any intervention or clinical trial + learning used in our models was supervised (i.e status re myeloid neoplasia had to be known). |

# Reporting for specific materials, systems and methods

We require information from authors about some types of materials, experimental systems and methods used in many studies. Here, indicate whether each material, system or method listed is relevant to your study. If you are not sure if a list item applies to your research, read the appropriate section before selecting a response.

## Materials & experimental systems

| n/a | Involved in the study |
|-----|----------------------|
| ☒ ☐ | Antibodies |
| ☒ ☐ | Eukaryotic cell lines |
| ☒ ☐ | Palaeontology and archaeology |
| ☒ ☐ | Animals and other organisms |
| ☐ ☒ | Clinical data |
| ☒ ☐ | Dual use research of concern |
| ☒ ☐ | Plants |

## Methods

| n/a | Involved in the study |
|-----|----------------------|
| ☒ ☐ | ChIP-seq |
| ☒ ☐ | Flow cytometry |
| ☒ ☐ | MRI-based neuroimaging |

# Clinical data

Policy information about clinical studies

All manuscripts should comply with the ICMJE guidelines for publication of clinical research and a completed CONSORT checklist must be included with all submissions.

| | |
|---|---|
| Clinical trial registration | N/A |
| Study protocol | N/A |
| Data collection | Clinical outcome data on myeloid neoplasia development was collected by the UK Biobank and by Drs Malcovati (Pavia CCUS cohort) and Drs Cargo & Smith (Leeds cohort) |
| Outcomes | Clinical outcome data on myeloid neoplasia development was collected by the UK Biobank and by Drs Malcovati (Pavia CCUS cohort) and Drs Cargo & Smith (Leeds cohort) |

