## [Peer Review File · Nature Genetics]

Peer Review Information

Manuscript Title: Multiparameter prediction of myeloid neoplasia risk

Corresponding author name(s): Dr Pedro Quiros and Professor George (S.) Vassiliou

Reviewer Comments & Decisions:

Decision Letter, initial version:

2nd Mar 2023

Dear Professor Vassiliou,

Your Article, "Multiparameter prediction of myeloid neoplasia risk" has now been seen by 3 referees - Reviewer #1 submitted their report yesterday. You will see from the comments below that while they find your work of interest, some important points are raised. We are interested in the possibility of publishing your study in Nature Genetics, but would like to consider your response to these concerns in the form of a revised manuscript before we make a final decision on publication.

We therefore invite you to revise your manuscript taking into account all reviewer comments. Please highlight all changes in the manuscript text file. At this stage we will need you to upload a copy of the manuscript in MS Word .docx or similar editable format.

*2) If you have not done so already please begin to revise your manuscript so that it conforms to our Article format instructions, available

[here](http://www.nature.com/ng/authors/article_types/index.html).

*3) Include a revised version of any required Reporting Summary: <https://www.nature.com/documents/nr-reporting-summary.pdf>

[redacted]

We hope to receive your revised manuscript within four to eight weeks. If you cannot send it within this time, please let us know.

Sincerely,

Safia Danovi
Editor
Nature Genetics

Referee expertise:

Referee #1: MPN, clinical

Referee #2: MPN, clinical

Referee #3: CH, cancer risk

Reviewers' Comments:

Reviewer #1:

Remarks to the Author:

In this study the authors report a further analysis of UK Biobank data with a focus on clonal hematopoiesis (CH) and risk of development of myeloid malignancy. The novelty of the study as articulated by the authors is that (1) their analysis is focused on other myeloid neoplasm subtypes in addition to acute myeloid leukaemia, (2) they incorporate haematological and biochemistry blood parameters and (3) they develop a risk tool (MPN-Predict) for clinical implementation.

It is my view that the “proof of principle that individuals that develop any MN subtype can be identified years in advance” is already rather well established and this does not represent a major step forward in the field. Previous studies have analysed haematological parameters alongside CH mutations in the same UKB cohort across all haematological malignancies (e.g. DOI: 10.1038/s41591-021-01521-4).

A personalised risk score for CH patients is potentially quite impactful and is novel, but as there are features of the UK Biobank that might introduce certain biases, this tool requires validation in large independent cohorts of patients (rather than quasi-randomisation) before clinical implementation.

I have some specific comments.

1. The methods state that VAF had to be >0.1 for predictive models yet many of the figures show pts with VAF <0.1 . Often an arbitrary cut off of VAF 2% is used to define CHIP. Can the authors carefully clarify exactly what VAF cut-off was used and how this relates to the VAF shown in the figures. Perhaps in some cases they refer to VAF as a % and in other cases as a proportion? The variant calling should be carefully benchmarked against published approaches. Both VAF and variant calling for UK biobank are described in a recent paper <https://doi.org/10.1182/blood.2022018825> where the authors highlight potential sequencing artifacts. Indeed, the incidence of CHIP was 1% lower (in the same dataset) so the authors need to carefully address this.
2. Of the 648 patients with a previous diagnosis of myeloid neoplasm (line 89), were these included in the 21362 individuals described with CH in the previous paragraph (line 83)?
3. The incidence of JAK2 CHIP is rather low compared to studies using targeted analysis for JAK2V617F e.g. DOI: 10.1182/blood.2019001113 in a Danish population. This may relate to VAF sensitivity? This Danish cohort is notable as many of the patients had hematological parameters consistent with a diagnosis of MPN (without a formal MPN diagnosis). This raises a broader concern that patients sampled

may have had a MN at the time of sampling (rather than CH) and this MBN was simply undiagnosed. How did the authors address this? Are persons with abnormal blood parameters in MPN or MDS range excluded from subsequent analysis? Clinically many patients have abnormal blood parameters dating back a number of years before diagnosis of MPN.

4. Again, specifically focusing on JAK2, the authors report that this occurs as a CH mutation with a prevalence of 1.9% (Fig 1A). In Fig 3F, the MPN-free survival associated with JAK2 mutation is between 80% and 50% after about 5 years follow up. This seems to predict an extremely high mortality and/or incidence of MPN and would predict many more MPN patients than occur in reality (incidence approximately 1/100,000 per year).

5. Related to this, it is unexpected that the incidence of MPN is so much higher than other myeloid neoplasms, almost double that of MDS or AML. What is the explanation for this?

6. Did MPN include CML (which is associated with CH)?

7. More granularity s required for the 129 of 2045 cases where multiple MN were diagnosed contemporaneously. This is not a phenomenon I recognise clinically to occur with such frequency and does raise a concern about veracity of the data, an issue with these large biobanks.

8. Given the recent evidence that MPN (and other MN) develop over many decades, with a modest year-on-year fitness advantage, is it not a surprise that CH mutations were only found in 32.7% pre-MN cases, typically <10 years before diagnosis). Can the authors model this somehow using data from the literature.

9. Do the authors have any data on the mutations and VAF at time of diagnosis on MN?

10. It is very difficult to see trends in blood cell parameters in different MN subgroups in Supp Fig 3 due to the normalisations used. It would be easier if blood parameters were shown according to usual units.

11. I was surprised that certain chromosomal abnormalities, strongly associated with myeloid malignancy such as 5q- or 9pLOH did not refine their model. What is the explanation for this?

12. Minor point, the panel in Fig 1 states that lasso regression is used to smooth curves in D & E (panel E does not show curves)

Reviewer #2:

Remarks to the Author:

The authors used UKBB data to identify individuals with CHIP and then developed a model to predict risk of progression from CHIP to myeloid neoplasia.

Major comment:

The authors have tested their MN-predict model only in healthy volunteers and as the authors note their data is therefore susceptible to “healthy volunteer bias”. In clinical practice, MN-predict will be most relevant to patients who are found to CHIP or CCUS and referred to a haematology or cardiology clinic for further evaluation. Can the authors validate the MN-predict model in “real life” clinically relevant cohort of individuals?

Figure-specific comments/questions:

Figure 1

Frequency of JAK2 = 1.9%, CALR = 0.6% indicates JAK2 mutation is approx three times more frequent than CALR mutation in healthy individuals. This finding does not align with previously published data of healthy individuals in the population where JAK2 mutation was found to be approx nineteen times more frequent than CALR mutation, using droplet digital PCR for genotyping: <https://pubmed.ncbi.nlm.nih.gov/31217187/>

How do the authors explain these discordant findings?

Do the authors think they are “missing” JAK2V617F mutations due to low JAK2 coverage (Supp Table 2)?

If so can they postulate how this impacts their findings?

Are the authors confident they are restricting CALR mutations to indel mutations in exon 9 that are associated with the development of MPN?

Also, what is the lower VAF limit the authors can detect and does it vary for different mutations (I could not find this stated in the manuscript)

Figure 2C

It is notable that the SRSF2 mutation is strongly associated with AML, MDS and MPN risk? Can the authors decipher additional factors that determine which MN subtype individuals with SRSF2 mutations develop?

Figure 3F

The authors have previously reported that JAK2V617F-mutant CH clones have the lowest fraction of clones growing at a constant rate, as compared to other CH-associated mutations (Fabre et al., Nature 2022). Yet in this paper they show that JAK2V617F is the mutation most strongly associated with MN development and that higher JAK2V617F VAF is associated with higher MPN risk. How do the authors align these somewhat contradictory findings?

Figure 3G – no relationship between MPL VAF and risk of developing MPN – can the authors explain?

It is notable that the MPL mutations that the authors include in their analyses include many non MPN-associated MPL mutations, some of which are typically germline (Supp Table 1). Can the authors re-do the analysis with MPN-associated MPL mutations only?

Since CALR is more common than MPL (Supp Figure 2), why is MPN risk for CALR mutations not shown?

Figure 4

Can the authors show the actual blood values rather than normalized values?

Is it possible that a subset of the individuals with elevated PLTs and HgB have undiagnosed MPN? Seeing the actual blood counts would be helpful in this regard.

Minor comments:

Figure 1 legend does not correspond to panels shown in figure – please correct

Figure 3A – the first gene

Reviewer #3:

Remarks to the Author:

The authors build a risk prediction tool for myeloid neoplasms using information on clonal hematopoiesis of indeterminate potential (CHIP) as well as several other clinical and demographic variables using data from the UKBiobank. The study was well-conducted and I was impressed with the predictive performance of their models. Generally, the story was easy to follow though some of the descriptions lacked sufficient detail. I have made several suggestions below for strengthening the paper.

Several parts of the Methods section were too brief and require much greater detail, perhaps as additional Supplementary Methods.

The models were run with a Cox PH model and predictive ability was measured via AUC. This is not so straightforward and the authors only mention that “ROC curves were constructed by comparing the probability of developing MN by the final time point with the real clinical outcomes, using the R package “pROC”.” Please expand on this description as I did not quite understand exactly how this was done. Did the authors use one of the definitions from Heagerty and Zheng (2005)? [Heagerty PJ, Zheng Y. Survival model predictive accuracy and ROC curves. *Biometrics*. 2005;61(1):92–105]. If so, which one? Perhaps an illustrative example would be helpful.

How was death as a competing risk handled? I assume that the authors did not run a Fine-Grey model with a cumulative incidence function, as this was not mentioned. Did they use a cause-specific hazard function? Please provide more details.

The description of the forward selection procedure requires more detail. I could piece together what the authors meant, but it took a while for me to look at Supp Table 5 in order to get it. Please explain this more carefully and in greater detail. Also, what metric of “concordance” was used in the forward selection procedure.

The authors should consider using a random survival forest or other ML technique to compare with the Cox PH model. Such methods are known to increase predictive performance.

Results

A comparison of the fit statistics in the training vs test sets would help the reader understand whether the model is overfit and a different model might be considered.

Is there any other risk tool that this can be compared to? If not, I would like to see a comparison of the final risk model with a model that includes the non-CHIP information, i.e., how much is the predictive performance enhanced by including information on CHIP?

I wasn't able to use the MN-predict hyperlink so I could not try it out the tool.

Can the authors provide an explanation/intuition for why the AUCs are so much better for MN in 0-1yr versus 1-5 and >5 years?

Supp Table 6 should display be HRs rather than the raw coefficients.

Author Rebuttal to Initial comments

We thank our reviewers for their positive assessments of our manuscript and for their helpful comments and suggestions. We have now responded to all comments in full by performing additional analyses, validating our models in two independent cohorts, generating new figures, adding new methodological detail and making several changes/additions to the text. The revised manuscript is significantly improved and its conclusions are very robust as a result.

Below is a point-by-point response to all reviewers' comments, with reference to the relevant changes/additions to the manuscript. The original reviewers' comments are in black font with our responses in blue. We use yellow highlight to point reviewers to changes in the manuscript text/figure/tables. In the manuscript text, we also use yellow highlight to indicate changes/additions.

Reviewers' Comments:

Reviewer #1:

Remarks to the Author:

In this study the authors report a further analysis of UK Biobank data with a focus on clonal hematopoiesis (CH) and risk of development of myeloid malignancy. The novelty of the study as articulated by the authors is that (1) their analysis is focused on other myeloid neoplasm subtypes in addition to acute myeloid leukaemia, (2) they incorporate haematological and biochemistry blood parameters and (3) they develop a risk tool (MPN-Predict) for clinical implementation.

It is my view that the "proof of principle that individuals that develop any MN subtype can be identified years in advance" is already rather well established and this does not represent a major step forward in the field. Previous studies have analysed haematological parameters alongside CH mutations in the same UKB cohort across all haematological malignancies (e.g. DOI: 10.1038/s41591-021-01521-4).

A personalised risk score for CH patients is potentially quite impactful and is novel, but as there are features of the UK Biobank that might introduce certain biases, this tool requires validation in large independent cohorts of patients (rather than quasi-randomisation) before clinical implementation.

We thank reviewer #1 for their positive overall assessment of our manuscript. We agree that previous studies have demonstrated that MN as a group can be identified years in advance. We wanted to make the point that this had not been demonstrated for individual MN subtypes, as this is pertinent to our

manuscript. However, to avoid misunderstanding, we have changed the relevant text in our abstract to remove the term “proof-of-principle”. The relevant sentence now reads as follows: ***“Our study demonstrates that individuals that develop any MN subtype can be identified years in advance...”*** (Abstract: line 47).

We also accept that UK Biobank (UKB) participants are healthier than the general population, something that can influence the findings of association studies of clonal hematopoiesis (CH) with non-haematological diseases. However, there is little evidence that the risk of MN is significantly influenced by lifestyle choices, with the possible exception of tobacco smoking, which is adequately represented in the UKB. Nevertheless, we entirely accept that validation of our findings in external cohorts would strengthen our work. However, large well-annotated studies equivalent to the UKB are lacking and even if they were available could be subject to healthy volunteer biases. Instead, we have chosen to validate our models in two real-world clinical cohorts of patients with clonal cytopenia of undetermined significance (CCUS) and show that our models perform very well in both (added Introduction: line 77-79, Results: line 214-231, Supplementary Tables 8 & 9 and Supplementary Figures 14 & 15). As these cohorts are quite different in composition to the UKB, this demonstrates the robustness of our models in different and clinically relevant contexts.

I have some specific comments.

1. The methods state that VAF had to be >0.1 for predictive models yet many of the figures show pts with VAF <0.1 . Often an arbitrary cut off of VAF 2% is used to define CHIP. Can the authors carefully clarify exactly what VAF cut-off was used and how this relates to the VAF shown in the figures. Perhaps in some cases they refer to VAF as a % and in other cases as a proportion? The variant calling should be carefully benchmarked against published approaches. Both VAF and variant calling for UK biobank are described in a recent paper <https://doi.org/10.1182/blood.2022018825> [eur03.safelinks.protection.outlook.com] where the authors highlight potential sequencing artifacts. Indeed, the incidence of CHIP was 1% lower (in the same dataset) so the authors need to carefully address this.

We thank the reviewer for these excellent points. Applying a VAF cut-off is indeed an effective way to remove genomic noise and sequencing artefacts, in order to improve the accuracy of variant calling. However, we used a slightly different approach to tackle the same problem. Instead of using an arbitrary VAF cut-off, our approach relied on statistical methods that take the minimum coverage, recurrence and distribution of VAFs of pre-filtered calls to remove putative false positives. We have now revised our approach to align it to that used in the aforementioned recent paper (Vlasschaert et al, Blood 2023) and adopted their somatic driver definitions for the reasons discussed below. A cut-off of VAF >0.1 was only used for hotspot mutations (listed in Supplementary Table 3), as these were identified using Samtools mpileup that does not apply statistical filters. For these, we used stricter inclusion criteria (VAF >0.1 and ≥ 3 reads).

To understand the differences between our calling approach and that by Vlasschaert et al, we compared our CH mutation calling methods against theirs. Overall, both ours and Vlasschaert et al's methods addressed the same problems using similar approach with differences in details/cut-offs. Both methods used Mutect2 to call variants and Samtools to rescue the U2AF1 variants that are systematically missed due to an error in the human GRCh38 reference sequence. Also, we both applied a binomial test to the number of mutant reads, in order to remove more germline variants. However, Vlasschaert et al's

method had two advantages over our original method, namely that they tested candidate driver mutations identified in >20 people for: i) correlation with age and ii) association with a common genetic variant in the TERT promoter (rs7705526) which is associated with a higher risk of CH, in order to improve the quality/veracity of driver mutation calls. We therefore adopted their driver definition as outlined in their paper. (Methods: line 359-367)

Setting a minimum number of mutant reads for calling CH mutations, can lead to varying sensitivity/specificity on detecting CH drivers dependent on sequencing coverage. In order to use the optimal cut-off for our study, we next investigated which number of mutant reads in the Mutect2 output resulted in the best performance of our models. For this we, tested three thresholds, namely: ≥ 2 , ≥ 3 and ≥ 5 mutant reads. Expectedly, the ≥ 3 cut-off gave the most similar data to Vlasschaert et al's paper, as they also used this cut-off (Supplementary Table 4). However, despite following their mutation calling approach, we detected ~2,000 more *TET2* mutations that Vlasschaert et al (mutation in other genes agreed closely). Looking at *TET2*, we found that our number of mutations at each recurrent hotspot also agreed well with Vlasschaert et al. However, we also identified ~2,000 low-recurrence *TET2* mutations (each with ≤ 3 cases in the UKB), including 1437 nonsense mutations and 604 missense mutations in the *TET2* functional domains (p.1104-1481 and 1843-2002, excluding known germline/artefact - see Supplementary Table 1), which meet their driver definitions, but were not listed in the supplementary data by Vlasschaert et al. Importantly, both missense and nonsense low-recurrence *TET2* mutations collectively showed positive correlations with age, mirroring the behavior of known drivers (e.g. *TET2* I1873T) and in clear contrast with known germline mutants (e.g. *TET2* Y867H) – see Figure R1A), indicating that most were likely to be drivers. Notably, inclusion of these low-recurrence mutations did not affect predictive model performance (Figure R1B-E). Based on these reasons, we believe it is appropriate to include these low-recurrence *TET2* mutations as CH drivers.

Figure R1: Examination of the impact of low-recurrence *TET2* mutations.

(A) UKB participants were grouped into 2-year age bins in order to examine the age distribution of low recurrence *TET2* nonsense mutations and missense mutations in functional domains (as defined in Supplementary Table 1). In each age bin, the fraction of individuals with the depicted mutation(s) was calculated (as the proportion of the total number of individuals with that mutation in the UKB) and then normalized to the total number of individuals in that bin. A known driver (I1873T) and germline mutation (Y867H) are included as positive and negative controls. **(B)** Hazard ratios (HR) for different types of MN obtained from models trained without low-recurrence *TET2* mutations. HRs show no significant difference from the original models. **(C-E)** Exclusion of low-recurrence *TET2* mutations from our MN prediction models did not affect model performance as assessed by area under curve (AUC) of ROC curves for **(C)** AML, **(D)** MDS and **(E)** MPN.

We then built our models (including ones for stepwise regression for feature selection and the final models) on the mutations called using each of these cut-offs (≥ 2 , ≥ 3 and ≥ 5 mutant reads), and found that all 3 cut-offs gave similar model performances, but “ ≥ 2 mutant reads” gave higher concordance scores for the AML model compared to the other 2 cut-offs (Supplementary Figure 18). Based on these observations, we chose “ ≥ 2 mutant reads” as our final cut-off. This cut-off improved the AUC of ROC curves of AML prediction from 0.73 to 0.78 for the $>5y$ category (Figure 4A before vs after revision). The 0-1y AUC decreased from 0.93 to 0.88 but this is less of an impact on the overall predictive performance, due to the small number of samples in this time range ($n=11$).

In the manuscript, we now:

- refer to the use of Vlasschaert et al’s method in Results: lines 84-86,

- explain this in more detail in **Methods (Line 356-364)**,
- update **Supplementary Table 1** to match their driver definition
- add **Supplementary Table 4** to show mutation calls and VAFs
- add **Supplementary Figure 18** to show the effect of cut-offs on our model performance

In consequence of changing the mutation calling approach, we have now identified slightly more CH mutations in the UKB, which leads to many changes of numbers throughout the manuscript highlighted in **yellow**, as well as updates of majority of the results:

- **Figures: 1A,B,E, 2A-C, 3A-G, 4A-C and 5A-F**
- **Supplementary Figures: 1A-B, 2A-C, 5A-C, 6A-B, 7C-E, 8A-C, 9A-B, 12, 18**
- **Supplementary Table: 1, 4, 6, 7**

2. Of the 648 patients with a previous diagnosis of myeloid neoplasm (line 89), were these included in the 21362 individuals described with CH in the previous paragraph (line 83)?

The 648 individuals with a prior diagnosis of MN were included in the analysis of 454,340 UKB whole exome sequences searched for the presence of CH. Out of these 648 patients, we detected CH mutations in 233 and these were included in the 21,362 individuals (now 22,735 after revisions outlined in our response to Point 1) with CH driver mutations. However, all 648 were excluded from subsequent analyses and model development/testing. To make this clearer, we have now added this bracketed clause in the second paragraph of Results: **“(of whom 233 had CH driver mutations)”** **Result: line 97.**

3. The incidence of JAK2 CHIP is rather low compared to studies using targeted analysis for JAK2V617F e.g. DOI: 10.1182/blood.2019001113 in a Danish population. This may relate to VAF sensitivity? This Danish cohort is notable as many of the patients had hematological parameters consistent with a diagnosis of MPN (without a formal MPN diagnosis). This raises a broader concern that patients sampled may have had a MN at the time of sampling (rather than CH) and this MBN was simply undiagnosed. How did the authors address this? Are persons with abnormal blood parameters in MPN or MDS range excluded from subsequent analysis? Clinically many patients have abnormal blood parameters dating back a number of years before diagnosis of MPN.

As alluded to by the reviewer, it is highly likely that the lower prevalence of JAK2 V617F in our study, compared to the Danish population study by Cordua et al, is due to differences in assay sensitivity. In fact Cordua et al used digital droplet PCR, a highly sensitive method that detected JAK2 V617F down to a VAF 0.009%. In that study, 508 of 613 (83%) JAK2 V617F cases had VAF<1%, which is below the sensitivity of exome sequencing. We now discuss the study by Cordua et al in **Discussion: lines 296-300).**

As regards the possibility of undiagnosed MPN, we agree that this is a valid consideration and we thank the reviewer for pointing it out. Although a diagnosis of MPN cannot formally be reached without ruling out alternative reasons for a raised hemoglobin or platelet count, we do concede that many individuals in the UKB had undiagnosed MPN. For this reason, we isolated all UK participants that met the latest

diagnostic criteria¹ for Polycythemia Vera (PV) and Essential Thrombocythemia (ET), in particular we used the following criteria:

PV: JAK2 mutation + Hb >16.5g/dl (men) or Hb >16.0g/dl (women)

These are the less stringent cut-offs that normally require a bone marrow biopsy confirmation, but we apply them here to ensure that we exclude all individuals who had PV at the time of blood sampling.

ET: JAK2 or CALR or MPL mutation + Plts $\geq 450 \times 10^9/l$

The requirement for a BM biopsy was waved, in order to ensure that we exclude individuals who had ET at the time of blood sampling

A diagnosis of myelofibrosis (MFS) cannot be made on blood test results, but any individuals with MFS and high platelet counts will have been removed by the ET criteria above.

This led to the exclusion of 108 or 1000 individuals that were subsequently diagnosed with MPN, of which 26 had PV (HGB = 17.9 ± 1.43 g/dL and *JAK2-V617F* VAF = 0.38 ± 0.2 , mean \pm sd) and 82 had ET (PLT = 675 ± 225 $10^9/L$, mean \pm sd; 51 with *JAK2-V617F*, 25 with *CALR* and 6 with *MPL* mutation). We retrained our models on the remaining 892 individuals. Model performance was slightly worse for MPN, but still very good (see new Figure 4C).

Like MFS, a diagnosis of MDS cannot be made on blood test results and requires a bone marrow biopsy. In fact cytopenias like anemia are relatively common in the absence of MDS and, even in the presence of somatic mutations the diagnosis can often be CCUS. So whilst it is possible that some of the UKB participants may have had MDS, we could not identify them with any degree of confidence, so we could not apply the same approach as we used for PV and ET.

We have now described the removal of likely undiagnosed MPN cases from training/validation of our models in Result: line 97-102 and Methods: 388-391. This leads to many changes in the numbers of cases throughout the manuscript highlighted in yellow, and updates of majority of analyses:

- Figures: 1F, 2A-D, 3A,F,G, 4A-C,F, 5A-F
- Supplementary Figures: 3D, 4D, 5A-C, 6A-B, 7C-E, 8A-C, 9A-B, 12, 15, 16, 17, 18
- Supplementary Tables: 6, 7

4. Again, specifically focusing on JAK2, the authors report that this occurs as a CH mutation with a prevalence of 1.9% (Fig 1A). In Fig 3F, the MPN-free survival associated with JAK2 mutation is between 80% and 50% after about 5 years follow up. This seems to predict an extremely high mortality and/or incidence of MPN and would predict many more MPN patients than occur in reality (incidence approximately 1/100,000 per year).

We are grateful for this comment as we had similar considerations when analyzing our data. The reported incidence of MPN varies widely² and this may be because the registration of MPNs by cancer registries was very variable prior to the introduction of ICD-O-3. Before this, ICD10 was in use and the MPNs were classified as “D” codes (i.e. not considered as cancer and thus not routinely registered). So whilst many of the published studies estimate an incidence of 1 per 100,000, more up-to-date and specialist studies significantly higher incidence rates. For example, in a large European study, MPN

incidence was found to be 3.1/100,000 per year³ and in a UK “real-world” study it was found to be 6/100,000 per year⁴. Also, the median age of UK biobank participants was 58 years (i.e. higher than the median for population studies) at the time of blood sampling and the median follow up was 12.6 years (range 7.4-15.5 years). The annual incidence of MPN in this age range is significantly higher and a recent study found it to be 18.6/100,000 in those aged 70-80 years⁵. If we apply this number to the UKB, the expected number of cases of MPN over this period would be:

$$\text{Incidence} * \text{number of participants} * \text{years (of follow-up)} = 0.000186 \times 454,340 \times 12.6 = 1064$$

This number is close to the observed number of MPNs in our study (n=1000) and in a recent study investigating germline risk of MPN amongst a very similar, albeit not identical, UKB participant population (n=1086)⁶.

However, please note that the 1.9% in Figure 1A refers to the percentage of cases of CH that carried *JAK2* V617F (rather than the % of UKB participants carrying *JAK2* V617F). We have now made this clearer in the legend to Figure 1A: line 113.

5. Related to this, it is unexpected that the incidence of MPN is so much higher than other myeloid neoplasms, almost double that of MDS or AML. What is the explanation for this?

As discussed above, the likely explanation for this is that the incidence of MPN is variable and overall higher than previously realized.

6. Did MPN include CML (which is associated with CH)?

We did not include CML in our studies as this disease is directly linked to the acquisition of the BCR-ABL fusion gene and is outside the scope of our study.

7. More granularity is required for the 129 of 2045 cases where multiple MN were diagnosed contemporaneously. This is not a phenomenon I recognise clinically to occur with such frequency and does raise a concern about veracity of the data, an issue with these large biobanks.

We thank the reviewer for this comment. We looked carefully at these 129 MN cases and decided that it was safer to remove them as we could not be certain which of the MN subtypes (AML, MDS, MPN or CMML) arose first or if an overlap syndrome was present.

For clarity, 71 of the 129 cases were diagnosed with more than one MN within 35 days (0-35 days, mean 5.2 days). In 60/71 cases, AML was one of the two diagnoses, the other being MDS (n=37) or CMML (n=16) or MPN (n=10). In the remaining 11/71 cases MDS was listed in 10, MPN in 9 and CMML in 5. In 6/71 cases, three diagnoses were listed within 35 days. We believe that, in several of these cases, the patients had dysplasia with increased blasts at a % blasts that placed them at the MDS/AML or CMML/AML cut-off (10-20%). Other cases may have had an MPN/MDS overlap syndrome. As we could not be certain of the nature of these MNs and as our aim was to generate models that predict individual MNs separately, we opted to not include them into any of our three models (AML, MDS or MPN).

In the remaining 58 of 129 cases, the second MN diagnosis was made >35 days after the first (36-1839 days, mean 497 days). AML was the first diagnosis in all 58 cases and followed by MDS in 28 cases, MPN

in 8 and CMML in 28. In 6/58 cases, three diagnoses were listed (within 172-1386 days). It is probable that some of these individuals had AML and then a treatment-associated MN or a return to a pre-AML neoplasia (e.g. MDS, MPN or CMML) that was present subclinically prior to AML diagnosis. As we could not distinguish between these possibilities, we did not use these cases to train or validate our MN predictive models.

Overall, as our aim was to develop different predictive models for each of the main MN subtypes, we decide that omitting these cases was the safest path, particularly as their numbers are relatively small to substantial aid model performance.

As regards the veracity of MN diagnoses in the UKB, it is very reassuring that both the genetic mutations (Figure 2B) and the blood count results (Figure 4D-F and Supplementary Figure 3B-D), reflect those of the downstream MN subtype.

We now summarize the above in Methods under “Data acquisition” in Methods: line 336-341

8. Given the recent evidence that MPN (and other MN) develop over many decades, with a modest year-on-year fitness advantage, is it not a surprise that CH mutations were only found in 32.7% pre-MN cases, typically <10 years before diagnosis). Can the authors model this somehow using data from the literature.

We thank the reviewer for this interesting suggestion. We recently described the longitudinal behavior of smaller JAK2 V617F clones over 10-15 years and found these to expand less predictably than any other type of CH (Fabre et al, Nature 2022, <https://doi.org/10.1038/s41586-022-04785-z>)⁷. The unpredictable behavior of JAK2 V617F clones can also be inferred from the Danish study by Cordua et al, where the prevalence of small JAK2 V617F clones is significantly higher than that of larger clones, whilst the same comparison for CALR clones shows that a larger % of the total CALR clones expand to a large size (also discussed in response to the second comment by reviewer 2).

Our findings here suggest that larger clones (detectable by exome sequencing) expand more rapidly and predictably than smaller ones, but we are not aware of any other large dataset in the literature that includes sufficient numbers of larger clones for model development. We previously performed a “look-back” study of 12 individuals that developed JAK2 V617F positive MPN 4.5-15.2 years after donating a blood sample for unrelated reasons (donor registration) and found JAK2 V617F in 9 of 12 (McKerrel et al, Blood Adv 2017 <https://doi.org/10.1182/bloodadvances.2017007047>)⁸. The 3 of 12 lacking JAK2 V617F had all given their blood samples >12 years prior to diagnosis (12.1, 12,6 and 15.2 years). In this study, we used sensitive sequencing that could robustly detect JAK2-V617F at VAF≥0.008, so the findings indicate that despite the very long history of JAK2-V617F MPN, pre-MN clones can be too small to detect at >12 years before diagnosis (even with sensitive sequencing).

We discuss the implications of the study by Cordua et al in Discussion: line 296-300.

9. Do the authors have any data on the mutations and VAF at time of diagnosis on MN?

No, unfortunately we do not have any molecular information from the time of MN diagnosis, as such information was not returned to the UKB. However, we do provide these data for the 108 individuals with undiagnosed MPN (see response to Comment 3 above). Lines 96-101.

10. It is very difficult to see trends in blood cell parameters in different MN subgroups in Supp Fig 3 due to the normalisations used. It would be easier if blood parameters were shown according to usual units.

In supplementary Figure 3 we wanted to display all blood test parameters in a single plot, so had to use the same quantification scale for all variables and opted for ten quantiles (Q1-Q10). Color-coding shows enrichment/ depletion (expressed as Odds Ratios) in each of the ten quantiles for each of the parameters, allowing for a visualization of trends for each MN subtype. For example, platelet counts (PLT) are enriched in Q10 (high) for MPN, but not for AML or MDS, for which they are enriched in Q1 (low).

To make comparisons between MN categories more visually apparent, we have now used the same rank order of variables across all four MN categories in Supplementary Figure 3.

11. I was surprised that certain chromosomal abnormalities, strongly associated with myeloid malignancy such as 5q- or 9pLOH did not refine their model. What is the explanation for this?
Most of the

We thank the reviewer for this fair point. We did observe significant association between chromosomal copy number changes (mCAs) and MN risk. However our models' AUCs ROC curves did not improve significantly by incorporating mCA information most probably due to: **a)** a relatively low number of cases with mCAs and **b)** linear dependency between mCA and blood/biochemistry parameters especially for pre-MDS/MPN cases. In other words, the presence of mCA affected blood/biochemistry parameters, which therefore captured the increased risk associated with the mC A. In light of this and as mCAs are not routinely captured by standard diagnostic assays, we have not included mCA into our final models.

Nevertheless, in order to avoid readers thinking that mCAs are not associated with MN risk, we have now added two panels to Supplementary Figure 7A & B, displaying the significant associations between pre-AML cases and -5q, pre-MDS and -5q/4qLOH, and pre-MPN and 9pLOH/+9p/+9 in the UKB and have added this text to the Results: Lines 178-184.

12. Minor point, the panel in Fig 1 states that lasso regression is used to smooth curves in D & E (panel E does not show curves)

Thank you for spotting this. LASSO regression was use to smoothen curves C and D (rather than D and E). We have now corrected this in Figure 1 legend: line 115-119.

Reviewer #2:

Remarks to the Author:

The authors used UKBB data to identify individuals with CHIP and then developed a model to predict risk of progression from CHIP to myeloid neoplasia.

Major comment:

The authors have tested their MN-predict model only in healthy volunteers and as the authors note their data is therefore susceptible to "healthy volunteer bias". In clinical practice, MN-predict will be most relevant to patients who are found to CHIP or CCUS and referred to a haematology or cardiology clinic for further evaluation. Can the authors validate the MN-predict model in "real life" clinically relevant cohort of individuals?

We thank the reviewer for this valid comment. The UKB does have a healthy volunteer bias, in the sense that people joining the UKB were more likely to be health conscious and display reduced cardiovascular and some other risks. However, as the risk of *de novo* myeloid cancers is not significantly influenced by lifestyle, with the possible exception of tobacco smoking, there is no reason to consider that the UKB cohort differs substantially from the general population with regards to myeloid neoplasia risk. As such, we believe that our model will perform well when such individuals are referred to CHIP/CCUS clinics and agree that validation in real life cohorts is important and we now include validation of our models in two independent “real life” CCUS cohorts in the revised manuscript.

Specifically, we applied our MN-predict model to two independent CCUS cohorts: i) the “Leeds CCUS cohort” composed of 204 patients with CCUS recruited from 2014-2016 and followed up for up to 5.5 years (mean \pm sd = 3.0 \pm 1.7) and ii) the “Pavia CCUS cohort” composed of 312 patients with CCUS and followed up for up to 15.1 years (mean \pm sd of diagnosis = 4.4 \pm 3.6). We found that our models performed very well in predicting progression to AML or MDS in the Leeds cohort and MDS in the Pavia cohort (this cohort only had 2 cases of progression to AML, so we could not test our AML model here).

These findings provide robust support for the relevance of our models to real life cohorts/patients. The new cohorts and analyses are now included in the revised manuscript: **Introduction: line 77-79**, **Result: line 214-231**, **Discussion: line 305-308** and **Methods: line 446-464**. We also added **Supplementary Figures 13 & 14** and **Supplementary Table 8 & 9** to show these results.

Figure-specific comments/questions:

Figure 1

Frequency of JAK2 = 1.9%, CALR = 0.6% indicates JAK2 mutation is approx three times more frequent than CALR mutation in healthy individuals. This finding does not align with previously published data of healthy individuals in the population where JAK2 mutation was found to be approx nineteen times more frequent than CALR mutation, using droplet digital PCR for genotyping: <https://pubmed.ncbi.nlm.nih.gov/31217187/> [eur03.safelinks.protection.outlook.com]

How do the authors explain these discordant findings?

Do the authors think they are “missing” JAK2V617F mutations due to low JAK2 coverage (Supp Table 2)? If so can they postulate how this impacts their findings?

This is an interesting point. As highlighted in our response to Reviewer 1 Point 3, in the aforementioned 2019 study by Cordua et al⁹, 508 of 613 (83%) JAK2 V617F cases had VAF<1%, which is below the level of detection by whole exome sequencing. By contrast for CALR mutations, Cordua et al found that only 19/32 (59%) of CALR clones had a VAF<1%. Overall, the ratio of **JAK2 V617F:CALR** mutant cases changed significantly by clonal size:

1. VAF<1% 508:19 = 26.7:1
2. VAF1-10% 75:4 = 18.8:1
3. VAF>10% 30:9 = 3.33:1

As the reviewer suspected, this shows that small JAK2 V617F clones are much more abundant than small CALR-mutant clones, but the difference is much less pronounced for large clones. Our study used exome

sequencing, so was only able to identify larger clones for which the *JAK2-V617F:CALR* ratio was 3.03, which is close to the Cordua et al ratio for large clones.

It is also worth noting that Cordua et al acknowledge that they may have underestimated the *CALR* mutation rate, as their approach was only able to detect the common *CALR* mutations (type 1 and type 2), but not the less common/recurrent *CALR* mutations (collectively ~15-20% of all *CALR* mutations). This would have overestimated the *JAK2-V617F:CALR* ratio, which should have been lower than 19. Our approach detected all types of *CALR* mutations.

With regards to how “missing” small *JAK2-V617F* clones may impact our findings, we believe that the impact is likely to be small as the expansion of small *JAK2* clones is known to be less predictable (Fabre et al, Nature 2022 and also evident by the data of Cordua et al above). By contrast, clones that reach a large size are manifestly much more likely to progress to MPN. As the main aim of a model such as MN-predict is to identify individuals at high risk, the detection of larger clones should adequately capture most high-risk individuals. Also, in clinical practice, anyone with a small *JAK2-V617F* clone would be monitored and highlighted as high risk if/when their clone expands.

We have now discussed this in Discussion: line 296-300

Are the authors confident they are restricting *CALR* mutations to indel mutations in exon 9 that are associated with the development of MPN?

Yes, we only selected MPN-associated frameshift mutations of *CALR* in Exon 9 as described in Supplementary Table 1. Out of the 134 *CALR* mutations, 63 were L367Tfs (Type 1, 52bp deletion site) and 50 were K385Nfs (Type 2, 5bp insertion site) and 21 were other types of less common *CALR* exon 9 mutations. These relative frequencies are in keeping with the MPN literature. We have now added Supplementary Table 4 to make this and other mutation calls explicit.

Also, what is the lower VAF limit the authors can detect and does it vary for different mutations (I could not find this stated in the manuscript)

Our mutation calling was based on a combination of two pipelines: Mutect2 (used for all mutations) and Samtools mpileup (used specifically to identify lower VAF hotspot mutations). For the filtering of Mutect2 output, we examined the recurrence of the called variant and the distribution of VAFs across the cohort and retained calls that passed our one-sided exact binomial test (described in Methods: Line 361-364), to remove germline mutations. For the Samtools output of hotspot mutations, we used the relatively strict threshold of “≥3 reads and a VAF>0.1” as these calls did not pass Mutect2 filters. For both methods, the ability to identify mutations depended on the sequencing coverage of different genes/exons. So the lower limit of mutation VAF varied between different genes/positions. To make this clear, we now provide statistics of VAF distributions for recurrent mutations in Supplementary Table 4, as well as our summary of read coverage by gene in Supplementary Table 3.

Figure 2C

It is notable that the *SRSF2* mutation is strongly associated with AML, MDS and MPN risk? Can the authors decipher additional factors that determine which MN subtype individuals with *SRSF2* mutations develop?

This is an excellent observation by the reviewer as *SRSF2* is the gene represented in appreciable numbers in all three pre-MN groups (Figure 2B). With regards to MPN, we believe that cases of *SRSF2* CH that progressed to MPN are likely to have developed myelofibrosis and we previously showed that *SRSF2* mutations can be acquired before *JAK2* V617F in such cases⁸. With regards to AML vs MDS, our data suggests that *SRSF2/TET2* co-mutated cases were more likely to develop MDS (9/12) than AML (3/12) (see Fig 2B & Supplementary Figure 3), whilst *SRSF2/IDH2* co-mutated cases were more likely to develop AML (6/9) than MDS (3/9). These differences are captured by our predictive models, as are those that vary by blood count results (e.g. raised MCV or reduced platelets are more common amongst individuals that develop MDS than AML (Fig 4D-E). So whilst no individual predictor is specific to AML or MDS, a number of indicators make one or other more likely. We now allude to this in Discussion: line 283-285

Figure 3F

The authors have previously reported that *JAK2*V617F-mutant CH clones have the lowest fraction of clones growing at a constant rate, as compared to other CH-associated mutations (Fabre et al., Nature 2022). Yet in this paper they show that *JAK2*V617F is the mutation most strongly associated with MN development and that higher *JAK2*V617F VAF is associated with higher MPN risk. How do the authors align these somewhat contradictory findings?

This is an interesting point that we also discuss in response to Reviewer #1's comment 8. In short it is clear that small *JAK2* V617F clones behave less predictably than larger clones. Our current study identified mutations through exome sequencing, so could not identify small clones. For our longitudinal manuscript (Fabre et al, Nature 2022), we used deep sequencing and most of the clones identified were small (VAF<1%) at study entry, with only 3 of 12 clones exceeding a VAF of 10% during the 15 year follow-up period. By contrast, most clones identified in the UKB had a VAF >5% (Supplementary Table 4). This supports the premise that the behaviour of large clones is much more predictable/deterministic than large clones. A similar conclusion can be reached from the data by Cordua et al (see response to the second comment of this reviewer), with a very large number of small clones and a much smaller number of small clones. We have now discussed this in Discussion: line 296-300

Figure 3G – no relationship between *MPL* VAF and risk of developing MPN – can the authors explain?

It is notable that the *MPL* mutations that the authors include in their analyses include many non MPN-associated *MPL* mutations, some of which are typically germline (Supp Table 1). Can the authors re-do the analysis with MPN-associated *MPL* mutations only?

Since *CALR* is more common than *MPL* (Supp Figure 2), why is MPN risk for *CALR* mutations not shown?

We thank the reviewer for this comment that pertains to Figure 3G, which shows that small and large *MPL* clones have similar MPN-free survival. We apologize as this was the result of an error on our part, as we used the wrong VAF sizes in this particular plot. We have now replaced this panel with the equivalent plot for *CALR* mutations as per our reviewer's suggestion (Figure 3G). In any case, after removing *JAK2*, *MPL* or *CALR* cases with blood test results compatible with possible undiagnosed MPN (in response to reviewer #1's comment 3), the number of *MPL*-mutant cases was reduced to only 3, making a Kaplan-Meier curve inappropriate.

Figure 4

Can the authors show the actual blood values rather than normalized values?

Is it possible that a subset of the individuals with elevated PLTs and HgB have undiagnosed MPN? Seeing the actual blood counts would be helpful in this regard.

This is a valid point and we have now replaced normalized values with actual blood counts/units in Figure 4 D-F. Also, we have now removed 108 cases with probable undiagnosed MPN (see reviewer #1, comment 3).

Minor comments:

Figure 1 legend does not correspond to panels shown in figure – please correct

Thank you - we have now corrected this in Figure 1 legend: line 115-119.

Figure 3A – the first gene

The first parameter (VAF) in the upper panel of Figure 3A is a measure of clonal size irrespective of the mutated gene. We have now clarified this by changing “VAF” to “VAF of largest clone” in Figure 3A, and also modified Supplementary Figure 6 legend and Supplementary Figure 8 legend.

Reviewer #3:

Remarks to the Author:

The authors build a risk prediction tool for myeloid neoplasms using information on clonal hematopoiesis of indeterminate potential (CHIP) as well as several other clinical and demographic variables using data from the UKBiobank. The study was well-conducted and I was impressed with the predictive performance of their models. Generally, the story was easy to follow though some of the descriptions lacked sufficient detail. I have made several suggestions below for strengthening the paper.

Several parts of the Methods section were too brief and require much greater detail, perhaps as additional Supplementary Methods.

We thank our reviewer for these comments and for the suggestions below that have certainly helped us improve our manuscript. We have now provided methodological details in Methods, conducted new analyses and added several new supplementary figures.

The models were run with a Cox PH model and predictive ability was measured via AUC. This is not so straightforward and the authors only mention that “ROC curves were constructed by comparing the probability of developing MN by the final time point with the real clinical outcomes, using the R package “pROC.” Please expand on this description as I did not quite understand exactly how this was done. Did the authors use one of the definitions from Heagerty and Zheng (2005)? [Heagerty PJ, Zheng Y. Survival model predictive accuracy and ROC curves. *Biometrics*. 2005;61(1):92–105]. If so, which one? Perhaps an illustrative example would be helpful.

We thank the review for pointing out the lack of detail regarding the method. The method we used was the incident/dynamic method described in by Heagerty and Zhen (2005) to compute time-dependent ROC curves. The incident/dynamic method is more suitable for this analysis than the cumulative/static, cumulative/dynamic or incident/static methods because we are generating curves for different non-overlapping time intervals from diagnosis (0-1, 1-5 and >5 years) rather than curves “within x years”, where the periods would overlap (e.g. 0-1, 0-5, 0-15 years).

To clarify this, we now have modified Figure 4 legend: line 246-250 and provided more details in Methods: line 429-438.

How was death as a competing risk handled? I assume that the authors did not run a Fine-Grey model with a cumulative incidence function, as this was not mentioned. Did they use a cause-specific hazard function? Please provide more details.

The method we used applied a specific risk function for each cause. In the new revision we have added explanation in Methods: line 412-414.

The description of the forward selection procedure requires more detail. I could piece together what the authors meant, but it took a while for me to look at Supp Table 5 in order to get it. Please explain this more carefully and in greater detail. Also, what metric of “concordance” was used in the forward selection procedure.

We have now added a detailed description of how forward stepwise regression was performed to Methods: line 418-428. We have now also specified the type of concordance metric (C-index)¹⁰ we used for the stepwise regression Methods: line 425.

The authors should consider using a random survival forest or other ML technique to compare with the Cox PH model. Such methods are known to increase predictive performance.

We have now constructed random forest models with parameters covering different numbers of trees and node splits. Overall, these models perform well, but not better than our regression models. The probable reason for this is that prediction of MNs does not rely on complex interactions between variables. However, we are not claiming that Cox regression is out-performing random survival forest because we have only done feature selection on the Cox regression and we have not performed extensive parameter scan on the random survival forest. Nevertheless, we appreciate the suggestion and believe it is important to show that a decision tree-based approach is also effective in this context. Therefore, we provide results of random forest models in Results: Line 206-208, Supplementary Figure 10 and Discussion: line 301-304, as well as Methods: line 440-444.

Results

A comparison of the fit statistics in the training vs test sets would help the reader understand whether the model is overfit and a different model might be considered.

We have now added a comparison between ROC curves obtained from the training set and the validation set and showed no significant difference between the models’ performance (Supplementary

Figure 11) indicating that overfitting or underfitting is unlikely. We also discussed this in the manuscript Results: line 208-210.

Is there any other risk tool that this can be compared to? If not, I would like to see a comparison of the final risk model with a model that includes the non-CHIP information, i.e., how much is the predictive performance enhanced by including information on CHIP?

This is an excellent suggestion, as CHIP mutations are responsible for many of the blood test changes. We have now analyzed how model performance is influenced by the inclusion of information on CHIP (somatic mutations). Overall, models lacking mutation information performed reasonably well, but their performance (ROC) improved by the addition of this information (Figure R2, A vs B, C vs D and E vs F). This improvement was most significant for AML, a disease in which blood count changes appear late in its evolution (vs MPN and MDS). We also tested how well blood test data predicted MN risk in cases who lacked mutations. Expectedly, blood test results were less good at predicting future MN in this context. Taken together these observations suggest that mutations cause change in blood tests and these capture some of the risk associated with these mutations. However, the type and VAF of mutations are of additional predictive value and it is important to retain these in our models.

Figure R2: Comparison of models using blood and biochemistry parameters only (excluding mutational parameters) with original models that includes mutational parameters

ROC curves from Cox proportional hazard models for prediction of progression to MNs, computed from predicted 15-year probability of MN-free survival and the diagnosis within the 15-year follow-up period. Individuals having at least 1 CH mutations (Mut) and individuals having no CH mutation (NoMut) were plotted separately. AUC=area under curve. **(A)** AML model using genotype and blood/biochemistry parameters. **(B)** AML model using blood/biochemistry parameters only. **(C)** MDS model using genotype and blood/biochemistry parameters. **(D)** MDS model using blood/biochemistry parameters only. **(E)** MPN model using genotype and blood/biochemistry parameters. **(F)** MPN model using blood/biochemistry parameters only.

I wasn't able to use the MN-predict hyperlink so I could not try it out the tool.

We apologize for this. We have now provided the URL instead of attaching a hyperlink in the manuscript text. **Results: line 256** and **Data availability: line 481**. You can also follow the URL here: <https://muxingu.shinyapps.io/webapp>

Can the authors provide an explanation/intuition for why the AUCs are so much better for MN in 0-1yr versus 1-5 and >5 years?

This is an excellent comment that probably alludes to the biology of how CH develops towards a MN:

As CH expands in size to represent a larger proportion of circulating blood cells, it begins to impose its own characteristics on the blood count. These characteristics are mutation-dependent, for example *SRSF2* and *SF3B1* mutations may lead to lower HGB, raised MCV and reduced platelet counts (reflected in the pre-MDS changes in Figure 4E), whilst *JAK2*, *CALR* and *MPL* clones lead to a raised platelet and/or hemoglobin values, reflected in the pre-MPN blood counts (Figure 4F). By contrast, most common pre-AML mutations (e.g. *DNMT3A*, *TET2* and *ASXL1*) have a lesser impact on blood count results (reflected in the pre-AML counts in Figure 4D). As we get nearer to the MN diagnosis, clones become bigger and so does their impact on blood count parameters. In other words, both the VAF size and the FBC changes become more pronounced closer to the diagnosis.

We believe that the reason for the improvement in AUC as we move closer to the diagnosis, is that larger clones have a more deterministic behavior, both in terms of their clonal expansion trajectories and the likelihood of acquisition of additional mutations (that are usually required to engender diagnoses such as AML and MDS). The deterministic behavior is best illustrated by *JAK2* V617F clones that behave unpredictably when they are small, but become a lot more predictable when they increase in size (see responses to reviewers #1 & #2 pertaining to this). Evidence for increased mutation acquisition is reflected in the fact that many cases of pre-MN have multiple driver mutations, whilst the majority of CH cases in UKB have a single mutation. It is also probable that having multiple mutations has an exacerbated impact on blood counts, increasing its effect on the AUC.

Another factor behind the improved AUC nearer to diagnosis may be that normal variation may mask subtle changes in blood counts that occur when CH clones are still small (a very large number of samples would overcome this).

We now comment on this in **Discussion: line 296-300**.

Supp Table 6 should display be HRs rather than the raw coefficients.

Thank you – we have corrected this in the **Supplementary Table 7**.

References

1. Arber, D.A. *et al.* International Consensus Classification of Myeloid Neoplasms and Acute Leukemias: integrating morphologic, clinical, and genomic data. *Blood* **140**, 1200-1228 (2022).
2. Titmarsh, G.J. *et al.* How common are myeloproliferative neoplasms? A systematic review and meta-analysis. *Am J Hematol* **89**, 581-7 (2014).
3. Gatta, G. *et al.* Rare cancers are not so rare: the rare cancer burden in Europe. *Eur J Cancer* **47**, 2493-511 (2011).
4. Roman, E. *et al.* Myeloid malignancies in the real-world: Occurrence, progression and survival in the UK's population-based Haematological Malignancy Research Network 2004-15. *Cancer Epidemiol* **42**, 186-98 (2016).
5. Hultcrantz, M. *et al.* Incidence of myeloproliferative neoplasms - trends by subgroup and age in a population-based study in Sweden. *J Intern Med* **287**, 448-454 (2020).
6. Bao, E.L. *et al.* Inherited myeloproliferative neoplasm risk affects haematopoietic stem cells. *Nature* **586**, 769-775 (2020).
7. Fabre, M.A. *et al.* The longitudinal dynamics and natural history of clonal haematopoiesis. *Nature* **606**, 335-342 (2022).
8. McKerrell, T. *et al.* JAK2 V617F hematopoietic clones are present several years prior to MPN diagnosis and follow different expansion kinetics. *Blood Adv* **1**, 968-971 (2017).
9. Cordua, S. *et al.* Prevalence and phenotypes of JAK2 V617F and calreticulin mutations in a Danish general population. *Blood* **134**, 469-479 (2019).
10. Harrell, F.E., Jr., Califf, R.M., Pryor, D.B., Lee, K.L. & Rosati, R.A. Evaluating the yield of medical tests. *JAMA* **247**, 2543-6 (1982).

Decision Letter, first revision:

16th Jun 2023

Dear Dr. Vassiliou,

Thank you for submitting your revised manuscript "Multiparameter prediction of myeloid neoplasia risk" (NG-A61755R). It has now been seen by the original referees and their comments are below. The reviewers find that the paper has improved in revision, and therefore we'll be happy in principle to publish it in Nature Genetics, pending minor revisions to satisfy the referees' final requests and to comply with our editorial and formatting guidelines.

Sincerely,

Safia Danovi
Editor
Nature Genetics

Reviewer #1 (Remarks to the Author):

The authors have carefully considered my previous comments and have made a number of quite major revisions to the paper as a result. I appreciate their efforts to carefully respond. The paper is significantly improved as a result of the revisions, also including additional revision made following the other reviewers' comments.

The authors will no doubt have noted that, in the interim, a paper has been published which analysed the same cohort of patients with the same aims (DOI: 10.1056/EVIDoa2200310). This paper describes an easy to apply risk score including integration of blood parameters for predicting risk of MN following CHIP or CCUS. The paper includes some analysis of risk of other events such as cardiovascular disease. There is no way around the fact that this substantially impacts on the novelty of the current work, although it could reasonably be argued that the two papers are essentially contemporaneous. Clearly the pros and cons of the model versus risk score in the respective papers needs to be addressed in detail (this will include some of the changes made in response to revisions which may not have been addressed in the NEJM Evidence paper e.g. inclusion of patients with MPN at the time of sampling). The authors should also explain the issue relating to U2AF1 mutation calling which was apparently not possible in the NEJM analysis using the same dataset.

The current paper predicts a very high incidence/prevalence of MPN, notwithstanding the arguments made in the rebuttal, this is much higher than I would expect for an age matched population included in UKB on the basis of published studies e.g. in the paper they reference, the incidence for all diagnosed MPNs was much lower at 4.45/100000. It may be that their study reflects the ground truth i.e. prevalence of MPN is much higher than hitherto appreciated and many pts with MPN are essentially undiagnosed for many years. This would be broadly in line with the Danish cohort. Some discussion relating to this should be added.

It is helpful that Fig 4 now shows standard units for blood parameters. This is quite informative as, despite the revisions, many patients included in the cohort still have quite abnormal blood counts at the time of sampling, including thrombocytosis, thrombocytopenia, anaemia, raised MCV. This is particularly the case for patients diagnosed <1 year. I have read the arguments made in the rebuttal, but it seems to be there is no need for a risk score for many of these patients as they would typically at the time of testing immediately be referred to haematology and a diagnosis of MN made. What the UKB data tell us is that there very likely many undiagnosed MN patients (including both MDS & MPN) in addition to the CHIP/CCUS pts in the cohort who subsequently develop MN at a later time. This should be made clear in the manuscript.

In the abstract I would suggest to add the word sometimes (or similar): “.....our study demonstrates that individuals that develop any MN subtype can SOMETIMES be identified years....” Otherwise the implication is that MN can always be identified years in advance which I do not believe is the case.

Overall I congratulate the authors on an excellent study, I tried using the portal they have developed and it is very informative and will be helpful to apply clinically.

Reviewer #2 (Remarks to the Author):

Thank you for your responses. No further questions.

One comment/suggestion: In the time since this manuscript was submitted, the clonal hematopoiesis risk score (CHRS) has been published: <https://evidence.nejm.org/doi/full/10.1056/EVIDoa2200310>

Since this used the same UKBB data and apparently the same Pavia CCUS validation cohort, it would be appropriate to reference this publication in the discussion.

Reviewer #3 (Remarks to the Author):

The authors have responded to all of my comments. I have no further edits or recommendations.

Final Decision Letter:

11th Jul 2023

Dear George,

I am delighted to say that your manuscript "Multiparameter prediction of myeloid neoplasia risk" has been accepted for publication in an upcoming issue of Nature Genetics.

Your paper will be published online after we receive your corrections and will appear in print in the next available issue. You can find out your date of online publication by contacting the Nature Press Office (press@nature.com) after sending your e-proof corrections. Now is the time to inform your Public Relations or Press Office about your paper, as they might be interested in promoting its publication. This will allow them time to prepare an accurate and satisfactory press release. Include your manuscript tracking number (NG-A61755R1) and the name of the journal, which they will need when they contact our Press Office.

Please note that *Nature Genetics* is a Transformative Journal (TJ). Authors may publish their research with us through the traditional subscription access route or make their paper immediately open access through payment of an article-processing charge (APC). Authors will not be required to make a final decision about access to their article until it has been accepted. [Find out more about Transformative Journals](https://www.springernature.com/gp/open-research/transformative-journals)

Authors may need to take specific actions to achieve [compliance](https://www.springernature.com/gp/open-research/funding/policy-compliance-faqs) with funder and institutional open access mandates. If your research is supported by a funder that requires immediate open access (e.g. according to [Plan S principles](https://www.springernature.com/gp/open-research/plan-s-compliance)) then you should select the gold OA route, and we will direct you to the compliant route where possible. For authors selecting the subscription publication route, the journal's standard licensing terms will need to be accepted, including [self-archiving-and-license-to-publish](https://www.nature.com/nature-portfolio/editorial-policies/self-archiving-and-license-to-publish). Those licensing terms will supersede any other terms that the author or any third party may assert apply to any version of the manuscript.

Please note that Nature Portfolio offers an immediate open access option only for papers that were first submitted after 1 January, 2021.

If you have not already done so, we invite you to upload the step-by-step protocols used in this manuscript to the Protocols Exchange, part of our on-line web resource, natureprotocols.com. If you complete the upload by the time you receive your manuscript proofs, we can insert links in your article that lead directly to the protocol details. Your protocol will be made freely available upon publication of your paper. By participating in natureprotocols.com, you are enabling researchers to more readily reproduce or adapt the methodology you use. [Natureprotocols.com](http://natureprotocols.com) is fully searchable, providing your protocols and paper with increased utility and visibility. Please submit your protocol to <https://protocolexchange.researchsquare.com/>. After entering your [nature.com](http://www.nature.com) username and password you will need to enter your manuscript number (NG-A61755R1). Further information can be found at <https://www.nature.com/nature-portfolio/editorial-policies/reporting-standards#protocols>

Sincerely,

Safia Danovi
Editor
Nature Genetics